# Inhibition of glucose metabolism selectively targets autoreactive follicular helper T cells

Seung-Chul Choi [1], Anton A. Titov [1], Georges Abboud[1], Howard R. Seay [1], Todd M. Brusko [1], Derry C. Roopenian[2], Shahram Salek-Ardakani[1] & Laurence Morel[1]

Follicular helper T (T$_{FH}$) cells are expanded in systemic lupus erythematosus, where they are required to produce high affinity autoantibodies. Eliminating T$_{FH}$ cells would, however compromise the production of protective antibodies against viral and bacterial pathogens. Here we show that inhibiting glucose metabolism results in a drastic reduction of the frequency and number of T$_{FH}$ cells in lupus-prone mice. However, this inhibition has little effect on the production of T-cell-dependent antibodies following immunization with an exogenous antigen or on the frequency of virus-specific T$_{FH}$ cells induced by infection with influenza. In contrast, glutaminolysis inhibition reduces both immunization-induced and autoimmune T$_{FH}$ cells and humoral responses. Solute transporter gene signature suggests different glucose and amino acid fluxes between autoimmune T$_{FH}$ cells and exogenous antigen-specific T$_{FH}$ cells. Thus, blocking glucose metabolism may provide an effective therapeutic approach to treat systemic autoimmunity by eliminating autoreactive T$_{FH}$ cells while preserving protective immunity against pathogens.

---

[1] Department of Pathology, Immunology, and Laboratory Medicine, University of Florida, 1395 Center Drive, Gainesville, FL 32610, USA. [2] The Jackson Laboratory, 600 Main Street, Bar Harbor, ME 04609, USA. Correspondence and requests for materials should be addressed to L.M. (email: morel@ufl.edu)

The germinal center (GC) is the primary site of clonal expansion and affinity maturation for B cells through survival and selection signals provided by follicular helper CD4$^+$ T (T$_{FH}$) cells. GC-derived plasma cells produce high-affinity antibodies against pathogens or autoantigens[1]. Controlling T$_{FH}$ cell numbers is essential for the optimal affinity maturation in GC response: an insufficient T$_{FH}$ generation underlies impaired humoral immune responses in primary immunodeficiencies, while excessive generation of T$_{FH}$ cells allows the survival of low-affinity self-reactive clones, resulting in the production of autoantibodies[2]. Systemic lupus erythematosus (SLE) is characterized by class-switched high-affinity auto-antibodies, indicating GC involvement[3]. The frequency of T$_{FH}$ cells is expanded in all spontaneous mouse models of lupus and a high frequency of circulating T$_{FH}$ cells has been reported in multiple cohorts of SLE patients, which often correlated with disease severity[4]. Accordingly, interrupting T$_{FH}$ cell differentiation by blocking CD40-CD40L interactions[5,6] or IL-21[7–10] signaling, or by delivering miR-146a[11], improved disease outcomes in lupus-prone mice. Moreover, several drugs that have promising results in SLE patients reduce the number of circulating T$_{FH}$ cells[12–15].

The cytokines and transcription factors that regulate T cell differentiation reprogram the metabolism of naive CD4$^+$ T cells into effector subset-specific metabolic profiles, which provide regulatory checkpoints to fine-tune T cell differentiation and function[16]. CD4$^+$ T cells of lupus patients[17] and mouse models of lupus[18] present metabolic alterations, which include high mTOR complex 1 (mTORC1) activity, glycolysis and oxidative metabolism. In the B6.Sle1.Sle2.Sle3 (TC for triple congenic) model of lupus that shares more than 95% of its genome with the congenic C57BL/6 (B6) controls[19], inhibiting glycolysis with 2-deoxyglucose (2DG) and the mitochondrial electron transport chain with metformin normalizes T cell metabolism and reverses autoimmune pathology[20]. These findings were confirmed in NZB/W F1 and B6.lpr mice, two other models of lupus[20,21]. Importantly, the frequency and number of T$_{FH}$ cells as well as GC B cells were normalized by this dual treatment, suggesting the autoreactive expansion of T$_{FH}$ cells was dependent on either glycolysis or mitochondrial metabolism, or a combination of the two.

The understanding of the metabolic requirements of T$_{FH}$ cells has been lagging comparatively to other CD4$^+$ T cell effector subsets. T$_{FH}$ cells induced by LCMV Armstrong viral infection are metabolically quiescent as compared to T$_H$1 cells[22], with a low PI3K-AKT-mTORC1 activation and an overall decreased mitochondrial and glucose metabolisms. These results are consistent with the findings that Bcl6[23] and PD-1[24], both highly expressed by T$_{FH}$ cells, independently inhibit cellular metabolism including glycolysis in vitro. However, gene targeting showed that mTOR activation is required for homeostatic and immunization-induced T$_{FH}$ differentiation in vivo[25,26] by enhancing glycolysis[26]. Moreover, mTORC1 activation is associated to autoreactive T$_{FH}$ cell expansion by promoting the translation of Bcl6, the master regulator of T$_{FH}$ cell gene expression, in the Def6$^{tr/tr}$Swap70$^{-/-}$ DKO mice[27]. In the framework of these results obtained in different models with different approaches, the specific metabolic requirements of spontaneous lupus T$_{FH}$ cells to expand have not been characterized, and it is unclear whether they are similar to the metabolic requirements of T$_{FH}$ cells that are induced by exogenous antigens.

Here, we show that the inhibition of glycolysis reduces the expansion of autoreactive T$_{FH}$ cells in four lupus-prone models, but it has little effect on the production of T-dependent (TD) antigen (ag)-specific antibodies, or the production of influenza-specific T$_{FH}$ cells in either non-autoimmune B6 or lupus-prone

TC mice. In addition, spontaneous lupus T$_{FH}$ but not virus-specific T$_{FH}$ cells express low levels of amino acid transporters as compared to B6 T$_{FH}$ cells. Accordingly, glutaminolysis inhibition with the glutamine analog 6-Diazo-5-oxo-L-norleucine (DON) prevents the production of TD Ag-specific antibodies, and impairs spontaneous GCs. Overall, this study showed that high glucose utilization is a unique requirement of autoreactive T$_{FH}$ cells, whereas glutamine metabolism is used by all T$_{FH}$ cells. These opposite metabolic programs suggest that autoreactive and Ag-specific T$_{FH}$ cells are driven by different mechanisms, and imply that inhibiting glycolysis can uniquely target pathogenic autoreactive T$_{FH}$ cells while preserving protective immunity against pathogens.

## Results

**Expanded spontaneous T$_{FH}$ population in lupus mice.** CD4$^+$ T cells from anti-dsDNA IgG-producing lupus-prone TC mice showed an increased expression of the early activation marker CD69, as well as an accumulation of effector memory T (T$_{EM}$) and T$_{FH}$ cells (Fig. 1a–c) as compared to B6 controls. Lupus mice also showed a decreased frequency of CXCXR5$^+$PD1$^+$Bcl6$^+$Foxp3$^+$ T$_{FR}$ relative to T$_{FH}$ cells (Fig. 1c). Similar results were obtained in young TC mice before the production of anti-dsDNA IgG (Supplementary Fig. 1a-d), indicating that CD4$^+$ T cell activation and the expansion of T$_{EM}$ and T$_{FH}$ CD4$^+$ T cells precedes the manifestation of autoimmunity. mTORC1 activity measured by phosphorylation of ribosomal protein S6 (pS6) was increased in total TC CD4$^+$ T cells[20]. Here, pS6 expression was increased in naive CD4$^+$ T cells (T$_N$, CD4$^+$CD44$^-$) and T$_{FH}$ (CD4$^+$CD44$^+$PD-1$^{hi}$PSGL-1$^{lo}$) cells from autoantibody producing and pre-autoimmune TC as compared with B6 mice (Fig. 1d and Supplementary Fig. 1e). The increased mTORC1 activation in CD44-negative CD4$^+$ T cells in TC mice may be due a small but significant increased frequency of cells that have lost CD62L and/or gained CD69 expression, presumably indicating a pre-activation status (Supplementary Fig. 1f). High mTOR expression by TC T$_{FH}$ cells was confirmed by histology (Fig. 1e and Supplementary Fig. 1g): TC CD4$^+$ T cells inside GCs showed abundant mTOR staining as compared to B6 CD4$^+$ T cells, which expressed mTOR in the T cell zone. STAT3 activation induces Bcl6 expression and STAT3 deficiency impaired T$_{FH}$ cell differentiation[28–30]. However, IL-21-mediated STAT3 phosphorylation was significantly decreased in CD4$^+$ T cells from SLE patients[31]. Both young and aged TC mice showed a reduced pSTAT3 expression level in CD4$^+$CD44$^+$ T$_{Act}$ and T$_{FH}$ cells, but an increased frequency of the small population of pSTAT3$^+$ in CD44$^-$ T$_N$ cells (Supplementary Fig. 1h, i). TC and B6 mice showed a similar frequency of proliferating T$_{Act}$ and T$_{FH}$ cells, but the frequency of proliferating CD44$^-$ T$_N$ cells was higher in TC mice (Fig. 1f and Supplementary Fig. 1i). Consistent with the higher number of CD4$^+$ T cells in lupus mice, the number of proliferating cells was higher for all three subsets in TC mice (Fig. 1f and Supplementary Fig. 1j). Finally, the expression of Bcl2 at the protein (Fig. 1g) and transcript (Fig. 1h) levels was increased in TC T$_{FH}$ cells. In sum, lupus TC mice show an expansion of T$_{FH}$ cells that precedes the production of auto-antibodies, and that is associated with increased mTORC1 activity and Bcl2 expression, as well as decreased STAT3 activation. This suggests that the number of T$_{FH}$ cells expands by resistance to apoptosis, and possibly by an increased differentiation from CD44-negative-derived precursors following an increased proliferation and STAT3 signaling.

**Glycolysis inhibition normalizes lupus T$_{FH}$ cells.** The dual inhibition of glycolysis and mitochondrial metabolism by the

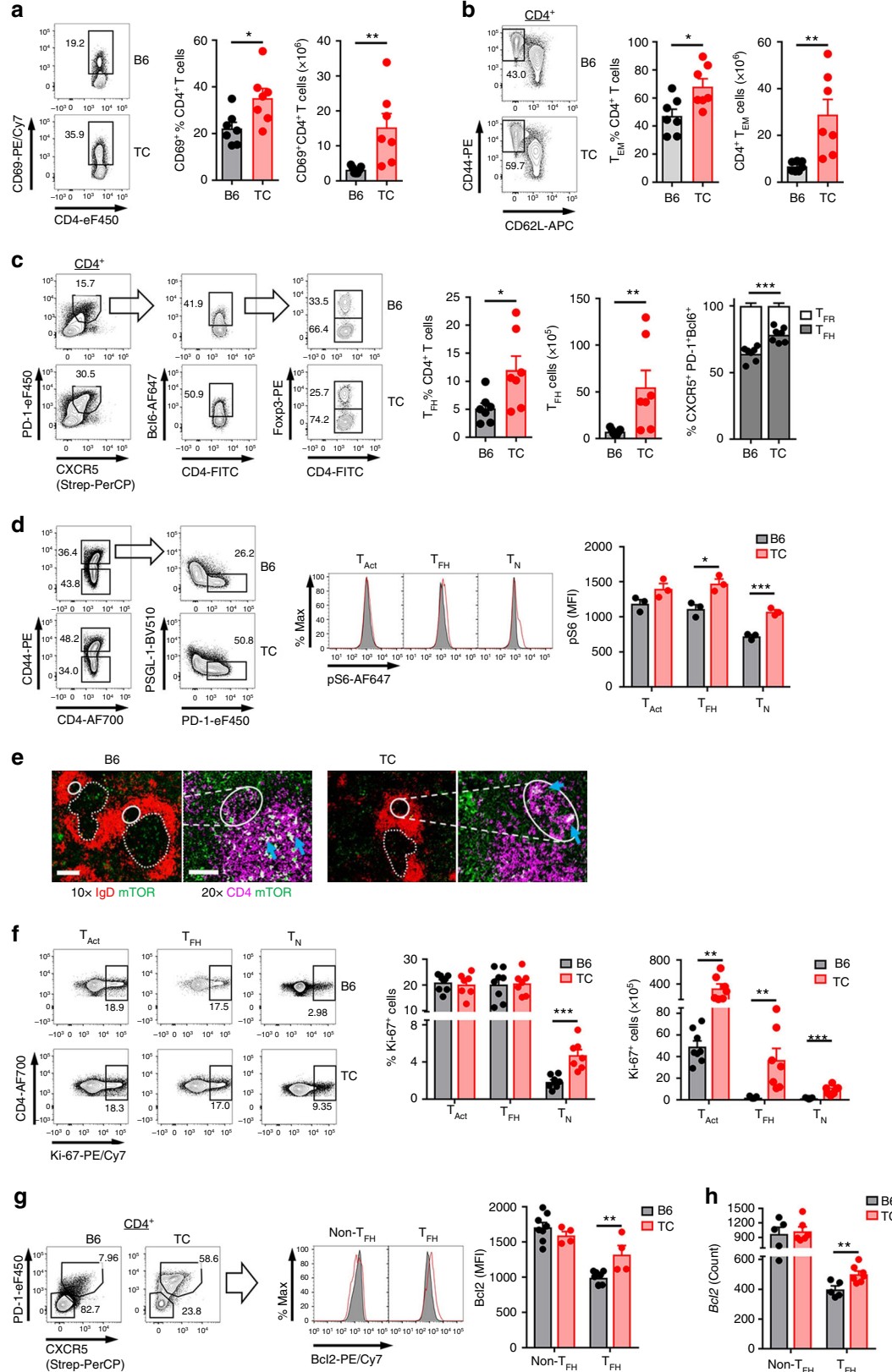

combination of 2DG and metformin reverses disease pathogenesis in TC, NZB/W F1[20], and B6.*lpr*[21] lupus-prone mice. This treatment significantly reduced anti-dsDNA IgG production, as well as the frequency of $T_{FH}$ and GC B cells in these three strains, as well as in a fourth model of lupus, the BXSB. *Yaa* mice, to

levels equivalent to that of B6 controls (Supplementary Fig. 2a). The 2DG plus metformin combination also reduced Bcl6 and PD-1 expression in total CD4+ T cells from these lupus strains (Supplementary Fig. 2b). Although treatment with the combination of these two metabolic inhibitors is required to suppress

**Fig. 1** Increased mTORC1 activity and survival in TC $T_{FH}$ cells. **a–c** Frequency and cell number of CD69$^+$ (**a**), CD4$^+$CD44$^+$CD62L$^-$ $T_{EM}$ cells (**b**), and CD4$^+$CXCXR5$^+$PD1$^+$Bcl6$^+$Foxp3$^-$ $T_{FH}$ (**c**) in CD4$^+$ T cells, with the relative distribution of $T_{FH}$ and Foxp3$^+$ $T_{FR}$ cells in CD4$^+$CXCXR5$^+$PD1$^+$Bcl6$^+$ cells. **d** Mean fluorescence intensity (MFI) of phospho-S6 in $T_N$ cells (CD4$^+$CD44$^-$), $T_{Act}$ cells (CD4$^+$CD44$^+$), and $T_{FH}$ cells (CD4$^+$CD44$^+$PD-1$^{hi}$PSGL-1$^{lo}$). **e** Representative mTOR staining (green) in the GC (plain line) and T cell zone (dashed line) relative to non-GC B cells (IgD$^+$ red) and CD4$^+$ T cells (purple). Arrows point to examples of mTOR staining. Scale bars: 200 μM for 10× and 10 μM for 20×. **f** Frequency and number of Ki-67$^+$ proliferating $T_N$, $T_{Act}$, and $T_{FH}$ cells. **g** Bcl2 MFI in CD4$^+$CXCXR5$^+$PD1$^+$ $T_{FH}$ and CD4$^+$CXCXR5$^-$PD1$^-$ non-$T_{FH}$ cells, with representative FACS plots on the left. **h** *Bcl2* mRNA expression in $T_{FH}$ and non-$T_{FH}$ cells as determined with a Nanostring custom panel. Spleens from 9-month-old B6 and TC mice, mean + s.e.m. of $N = 3$–8 mice per group compared with $t$ tests. *$P < 0.05$, **$P < 0.01$, and ***$P < 0.001$

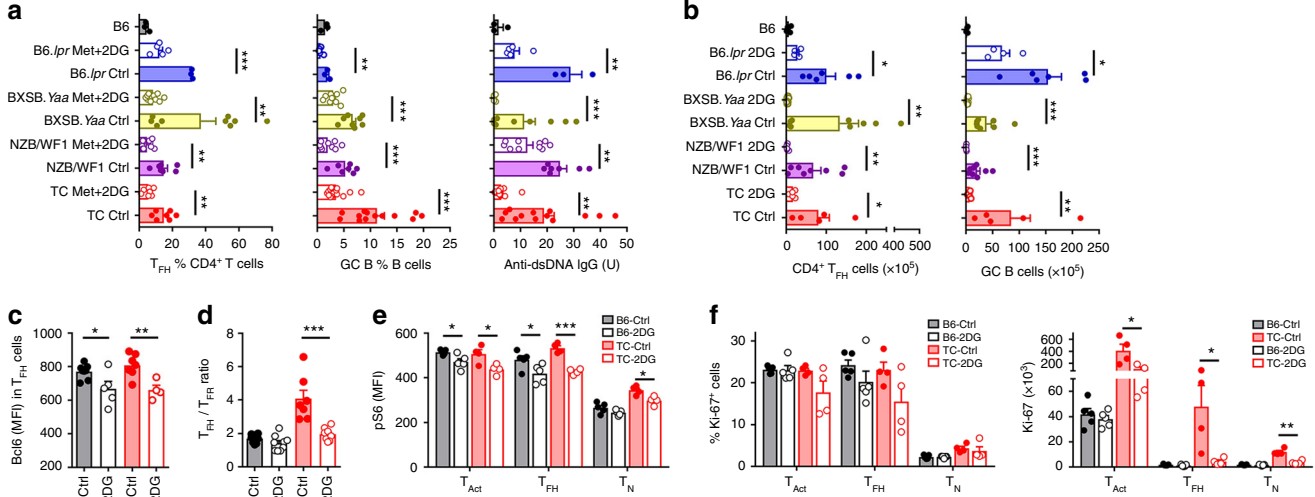

**Fig. 2** Autoreactive $T_{FH}$ cells are sensitive to glycolysis inhibition. Frequency (**a**) and cell number (**b**) of $T_{FH}$ cells (left, CD4$^+$CXCXR5$^+$PD1$^+$Bcl6$^+$Foxp3$^-$) in CD4$^+$ T cells and GC B cells (middle, B220$^+$GL7$^+$FAS$^+$) in B cells, as well as serum anti-dsDNA IgG (right) in four strains of lupus-prone mice treated with 2DG for 8 weeks starting at 5–6 months of age, as compared to untreated age-matched controls. Contemporary untreated B6 mice are shown as reference. Mean + s.e.m. of $N = 3$–28 mice per group. **c–f** 3–4-month-old B6 and TC mice were treated 2DG or not for 1 month. MFI of Bcl6 in $T_{FH}$ cells (**c**) and the ratio of $T_{FH}$ (CD4$^+$CXCXR5$^+$PD1$^+$Bcl6$^+$Foxp3$^-$) to $T_{FR}$ (CD4$^+$CXCXR5$^+$PD1$^+$Bcl6$^+$Foxp3$^+$) cells (**d**). Phospho-S6 (**e**) as well as frequency and number of Ki-67$^+$ (**f**) cells in $T_N$ cells (CD4$^+$ CD44$^-$), $T_{Act}$ cells (CD4$^+$CD44$^+$), and $T_{FH}$ cells (CD4$^+$CD44$^+$PD-1$^{hi}$PSGL-1$^{lo}$). Mean + s.e.m. of $N = 4$–9 spleens per group compared with $t$ tests. *$P < 0.05$, **$P < 0.01$, and ***$P < 0.001$

clinical disease in older mice, 2DG alone is sufficient to prevent disease development and autoantibody production[21]. To compare lupus-prone strains that have a different timing of autoimmune activation, we treated all mice starting at around 5 months of age when they were anti-dsDNA IgG positive but without clinical disease. After 8 weeks of treatment, 2DG reduced the production of anti-dsDNA IgG, as well as the frequency and number of $T_{FH}$ and GC B cells in these four lupus strains (Fig. 2a, b). 2DG also reduced Bcl6 expression in both B6 and TC $T_{FH}$ cells (Fig. 2c) and normalized the $T_{FH}/T_{FR}$ ratio in TC mice (Fig. 2d). The addition of metformin did not further reduce the frequency of $T_{FH}$ cells (Supplementary Fig. 2c), while 2DG alone has little effect on the frequency of $T_{EM}$ cells and CD69 expression in TC mice (Supplementary Fig. 2d, e). 2DG reduced the number of total splenic CD4$^+$ T cells in all four lupus-prone strains, but not in B6 mice (Supplementary Fig. 2f), which is consistent with CD4$^+$ T cells from lupus mice being glycolytic[20,21]. 2DG had a variable effect on CD4$^+$ T cell activation, decreasing the frequency of CD44$^+$CD4$^+$ T cells only in NZB/W F1 and BXSB.*Yaa* mice, but decreasing their number in all strains 2DG, except B6.*lpr* mice (Supplementary Fig. 2g). These results indicate that spontaneous lupus $T_{FH}$ cells are uniformly sensitive to the inhibition of glycolysis comparatively to other activated T cells that show a more variable response among strains. CD4$^+$ T cell effector differentiation depends on an increased glycolysis and mTORC1 activity[32]. Accordingly, 2DG reduced mTORC1 activation in both TC and B6 $T_{Act}$ and spontaneous $T_{FH}$ cells, as well as in TC $T_N$ cells

(Fig. 2e). In addition, 2DG markedly decreased the number of proliferating CD4$^+$ T cells in TC mice, including $T_{FH}$ cells (Fig. 2f). These results demonstrate that the expansion of spontaneous $T_{FH}$ and GC B cells and the resulting production of autoantibodies in lupus mice depend on glycolysis and mTORC1 activation.

**The TD humoral response does not require glycolysis.** To determine whether inhibiting glycolysis affects humoral immune responses to a TD exogenous antigen, mice were immunized with 4-hydroxy-3-nitrophenylacetyl conjugated to keyhole limpet hemocyanin (NP-KLH). We used young mice because aged TC mice respond poorly to TD immunization[33]. 2DG treatment started 2 weeks before immunization and continued for the duration of the experiment. The primary response was analyzed 10 d after immunization and the memory response 7 weeks after the first immunization and NP-KLH boosts at weeks 2 and 6. 2DG reduced the frequency and number of total TC GC B cells in both primary and memory responses (Fig. 3a). However, 2DG had no effect on the high frequency and number of TC plasma cells (Fig. 3b), but it reduced anti-dsDNA IgG levels in TC mice tested for the memory response (Fig. 3c). As previously reported[33], the frequency and number of NP-specific GC B cells was decreased in TC mice as compared with B6 controls in the primary response (Fig. 3d). More importantly, 2DG did not decrease the frequency and number of these NP-specific GC B cells (Fig. 3d). As for total plasma cells, the frequency and number of

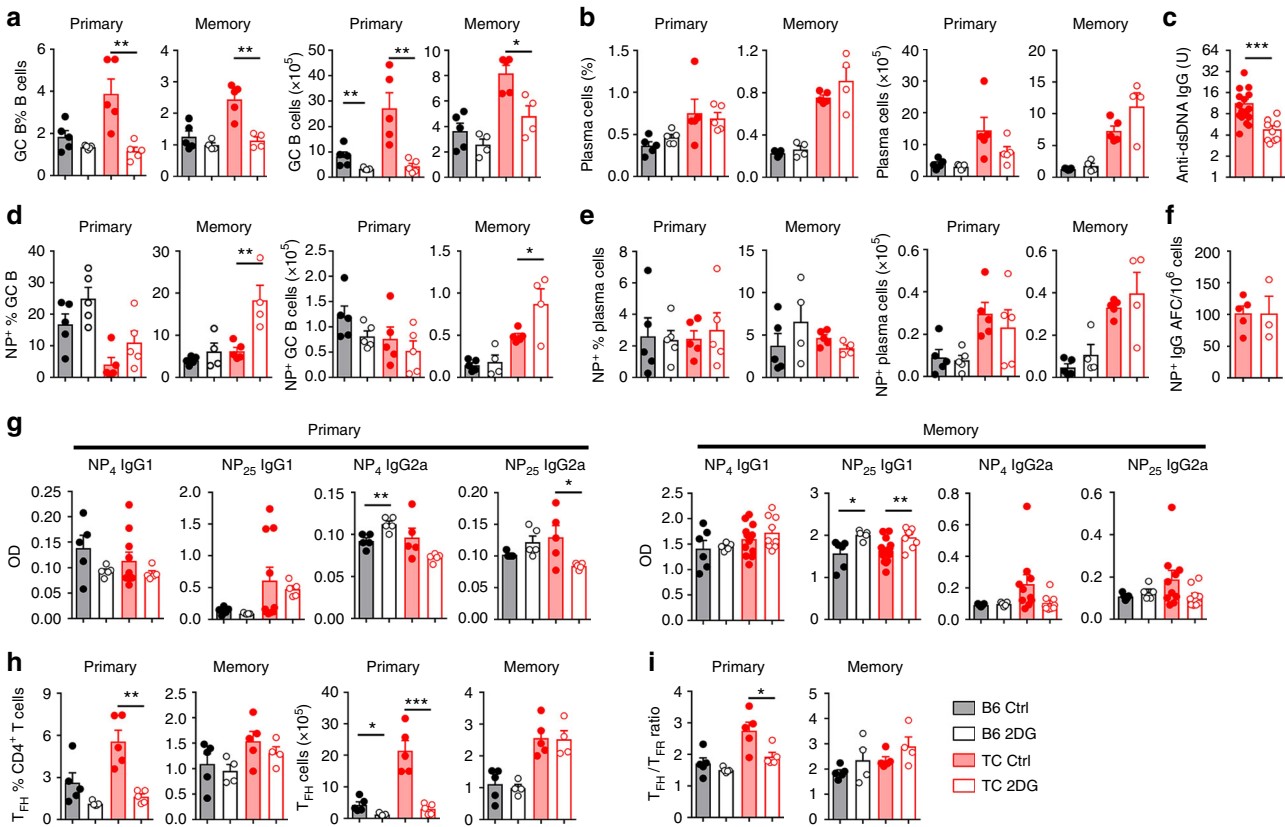

**Fig. 3** Glycolysis inhibition does not affect the TD-humoral response. After pre-treatment with 2DG for 2 weeks, 8–10-week-old B6 and TC mice were immunized with NP-KLH in alum and maintained under 2DG treatment until sacrifice. Mice were analyzed 10 d after immunization (primary), or 7 weeks after the first immunization following 2 boosts with same antigen 2 and 6 weeks after the primary immunization (memory). Frequency and number of total (**a**, **b**) or NP-specific (**d**, **e**) GC B cells and plasma cells (gating shown in supplementary Fig. 3). Serum anti-dsDNA IgG (**c**) and NP-specific IgG antibody-forming cells (**f**) in TC mice in the memory response. **g** Serum levels of high-affinity anti-NP$_4$ and low-affinity anti-NP$_{25}$ IgG1 and IgG2a in the primary (left) and memory (right) responses. Frequency and number of splenic $T_{FH}$ cells (**h**) and $T_{FH}/T_{FR}$ cell ratio (**i**) in the primary and memory responses. All cell analyses were performed with splenocytes. Mean + s.e.m. of $N = 5$ mice per group compared with $t$ tests. $*P < 0.05$, $**P < 0.01$, and $***P < 0.001$. For simplification, statistical differences between strains are not shown

NP$^+$ plasma cell was similar between 2DG-treated and control TC and B6 mice (Fig. 3e). Furthermore, 2DG has no effect on the number of NP-specific IgG producing cells in memory-immunized TC mice (Fig. 3f). The levels of serum NP-specific IgG1, which is the main isotype produced in this TD immunization, mirrored the cellular results, even with an increase of low-affinity NP-specific memory IgG1 (Fig. 3g). 2DG tended to decrease the level of $T_H1$-driven NP-specific IgG2a, only observed in the TC memory response (Fig. 3g). Overall, these results suggest that autoreactive GCs, but not GCs induced by exogenous Ag, are sensitive to glycolysis inhibition. The frequency and number of total $T_{FH}$ cells and the ratio of $T_{FH}$ to $T_{FR}$ cells were decreased in 2DG-treated TC mice in the primary, but not in the memory response (Fig. 3h, i), when the frequency of NP-KLH-specific $T_{FH}$ relative to spontaneous $T_{FH}$ cells is likely much higher than in the primary response. Importantly, B6 $T_{FH}$ cells, which contain only a small frequency of autoreactive cells, were not affected by 2DG, suggesting that glycolysis is not necessary for the differentiation of exogenous Ag-specific $T_{FH}$ cells.

**Exogenous Ag-specific $T_{FH}$ cells do not require glycolysis.** We tested the effect of 2DG on Ag-specific $T_{FH}$ cells induced by a non-lethal dose of influenza virus A/Puerto Rico/8/1934 H1N1 (PR8). Virus-specific CD4$^+$ T cells were detected with the MHC class II tetramer recognizing the nucleoprotein (NP) immunodominant viral epitope NP$_{311-325}$ (NP-Tet)[34,35] 10 d after

infection. Weight loss, a clinical indicator of influenza infection, was observed in B6 mice and was enhanced by 2DG, while neither 2DG-treated nor control TC mice lost weight (Supplementary Fig. 4a). 2DG did not affect the frequency of virus-specific CD4$^+$ T cells (Fig. 4a), and $T_{FH}$ cells (Fig. 4b and Supplementary Fig. 4b) in either strain, but the number of virus-specific $T_{FH}$ cells was lower in 2DG-treated TC mice than in controls (Fig. 4b), corresponding to a global effect of 2DG on lymphoid expansion[21]. Both the frequency and number of total TC $T_{FH}$ cells, which contain both spontaneous and virus-specific cells, were however decreased by 2DG (Fig. 4c), as we observed with non-immunized (Fig. 2a) and NP-KLH-immunized mice (Fig. 3h). Furthermore, the 2DG treatment did not decrease the amount of 3 isotypes of NP-specific IgG 4 weeks after infection (Fig. 4d), whereas it reduced anti-dsDNA IgG production in the same TC mice (Fig. 4e). A similar pS6 expression was observed in NP-Tet$^{pos}$ $T_{FH}$ cells between B6 and TC mice 10 d after infection (Supplementary Fig. 4c). Unexpectedly, glycolysis inhibition increased pS6 expression in NP-Tet$^{neg}$ $T_{FH}$ cells and there was a similar trend for NP-Tet$^{pos}$ $T_{FH}$ cells in TC mice. This result suggests that glycolysis is not the main pathway of mTORC1 activation during exogenous Ag-specific $T_{FH}$ cell differentiation. In addition, 2DG had no effect on STAT3 activation and proliferation in $T_{FH}$ cells (Supplementary Fig. 4d, e), except for a small increase in the proliferation of NP-Tet$^{neg}$ TC $T_{FH}$ cells (Supplementary Fig. 4e). Thus, these results strongly suggest that

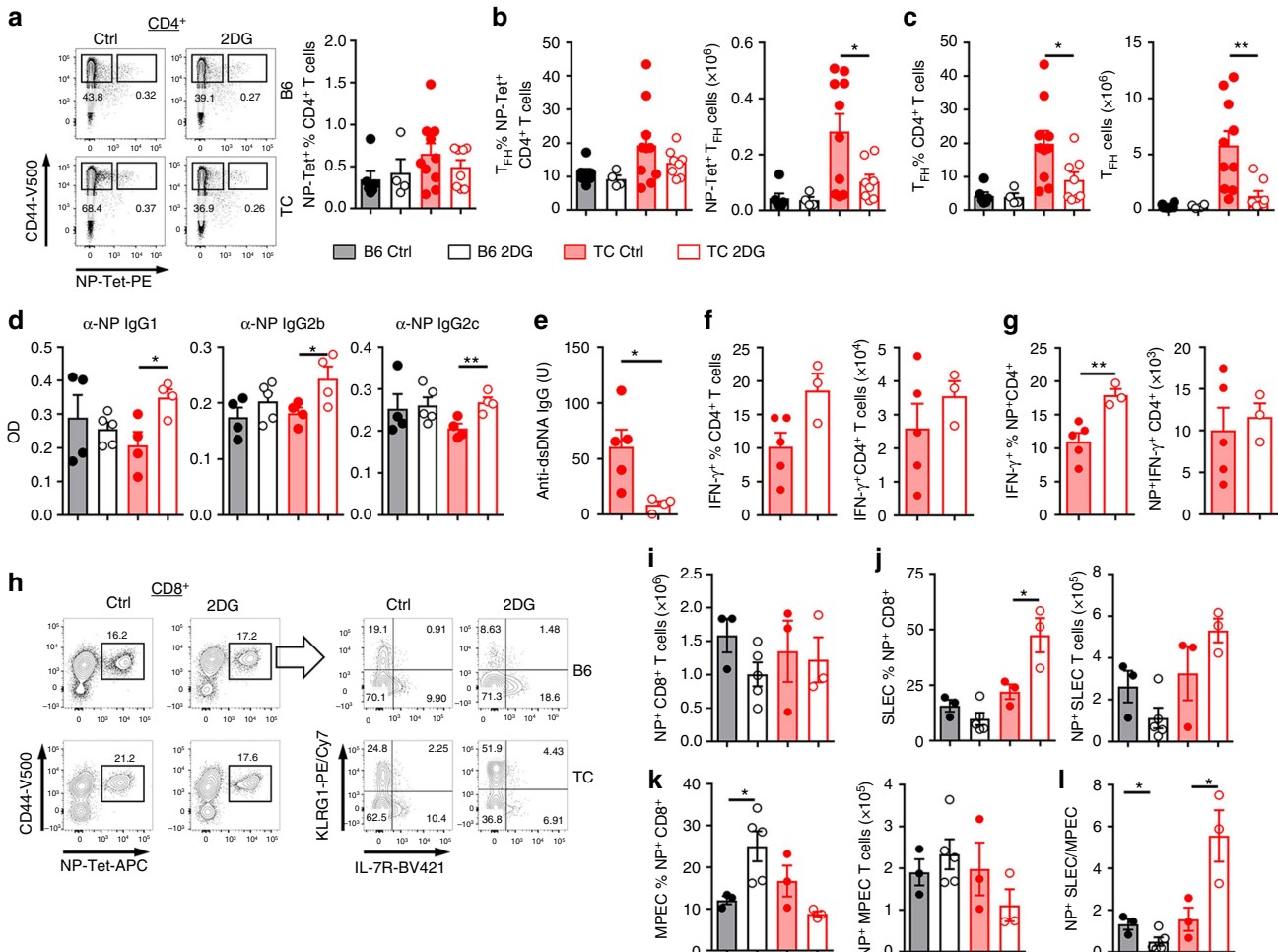

**Fig. 4** Glycolysis inhibition does not impair the humoral response to influenza immunization. 8–10-week-old B6 and TC mice were infected with PR8 influenza virus. 2DG treatment was initiated 2 weeks before infection and maintained until termination. CD4$^+$ and CD8$^+$ T cells were analyzed 10 d after infection. **a** Representative FACS plot (left) and percentage (right) of splenic influenza virus-specific CD4$^+$ T cells (CD4$^+$CD44$^+$NP-Tet$^{pos}$). **b, c** Frequency and number of CD4$^+$CD44$^+$PD-1$^{hi}$PSGL-1$^{lo}$ T$_{FH}$ cells in PR8 virus-specific NP-Tet$^{pos}$CD4$^+$ T cells (**b**) and in total CD4$^+$ T cells (**c**). Serum levels of PR8 NP-specific IgG1, IgG2b and IgG2c (**d**), and anti-dsDNA IgG (**e**) 30 d after low dose infection. **f, g** Frequency and number of IFNγ$^+$ in total CD4$^+$ T cells (**f**) and PR8 virus-specific NP-Tet$^{pos}$CD4$^+$ T cells (**g**) in the lung. **h, i** Analysis of NP-Tet$^{pos}$CD8$^+$ T cells in the lung: representative FACS plots (**h**), total number (**i**), frequency and number of SLEC (**j**) and MPEC (**k**) among these NP-Tet$^{pos}$CD8$^+$ T cells, and SLEC/MPEC ratio (**l**). Mean + s.e.m. of N = 3–8 mice per group compared with t tests. *P < 0.05 and **P < 0.01

glycolysis is not necessary for exogenous Ag-specific T$_{FH}$ cell differentiation.

We also examined the effect of 2DG on IFNγ-producing CD4$^+$ T cells, as well as virus-specific CD8$^+$ T cells in the lung, both of which being critical for protection[36]. The number and frequency of either total IFNγ$^+$ or virus specific NP-Tet$^{pos}$IFNγ$^+$ CD4$^+$ T cells were similar in the lung (Fig. 4f, g) and spleen (Supplementary Fig. 4f, g) between 2DG-treated and control TC mice. This confirms earlier results showing that metformin but not 2DG normalizes IFNγ production in TC mice[21]. Similarly, 2DG did not affect the number of NP-Tet$^{pos}$CD8$^+$ T cells in the lung (Fig. 4h, i) and spleen (Supplementary Fig. 4h) of either B6 or TC mice. Interestingly, 2DG had opposite outcomes on the type of virus-specific effector CD8$^+$ T cells: 2DG expanded short-lived effector cells (SLEC, KLRG1$^+$CD127$^-$NP-Tet$^{pos}$CD8$^+$) in TC mice while it expanded memory precursor effector cells (MPEC, KLRG1$^-$CD127$^+$NP-Tet$^{pos}$CD8$^+$) in B6 mice (Fig. 4i–k and Supplementary Fig. 4i, j) leading to opposite SLEC/MPEC ratios between the two strains in the lung (Fig. 4l) and a similar trend in the spleen (Supplementary Fig. 4k). These

results suggest that the inhibition of glycolysis does not prevent the differentiation of virus-specific protective CD4$^+$ and CD8$^+$ T cells, but modulates the type of effector cells it produces. Virus-induced differentiation of SLEC is favored by inflammatory signals such as CpG and IL-12[37], both of which have been associated with lupus. How this process is enhanced by the inhibition of glycolysis needs to be further explored.

**Differential expression of solute transporters in TC T$_{FH}$ cells.** Based on the observation that T$_{FH}$ cells were more sensitive to glycolysis inhibition than other types of activated CD4$^+$ T cells in TC mice (Supplementary Fig. 2c–e), we compared the expression of *Hif1a* and *Mct4*, two key genes in the glycolytic pathway, between total CD4$^+$ T cells and spontaneous T$_{FH}$ cells (CD4$^+$CXCR5$^+$PD-1$^+$) sorted from 8-10 month old TC mice (Supplementary Fig. 5a). TC T$_{FH}$ cells showed a 3–5 fold increased expression of *Hif1a* and *Mct4* (Fig. 5a), and an increased Hif1α protein expression (Fig. 5b) as compared to total CD4$^+$ T cells. Moreover, the extracellular acidification rate (ECAR), a measure of glycolysis, was higher in TC T$_{FH}$ than B6 T$_{FH}$ cells, while

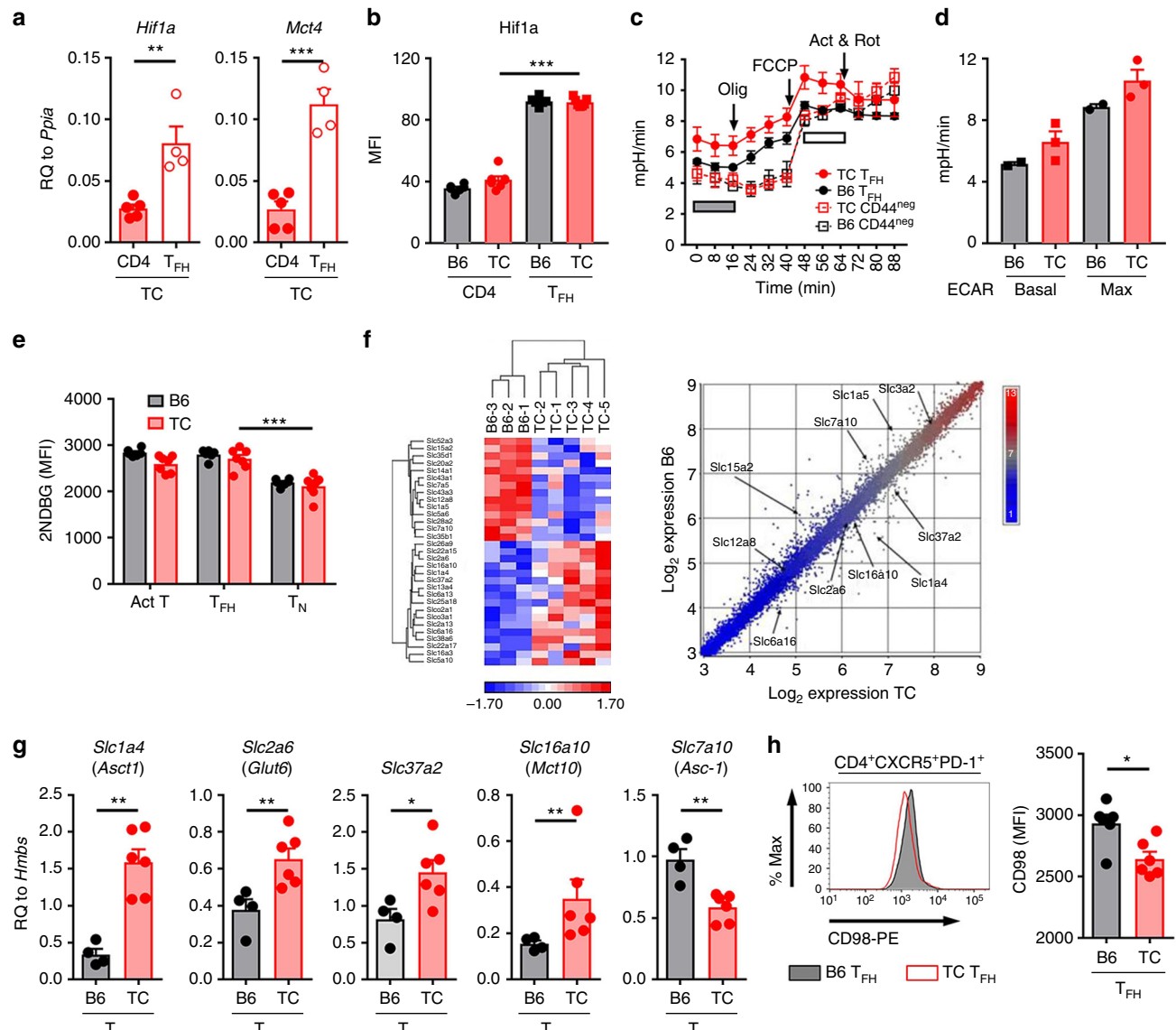

**Fig. 5** Spontaneous TC T$_{FH}$ cells are glycolytic and show an altered expression of solute transporters. **a** *Hif1a* and *Mct4* gene expression was compared between total CD4$^+$ T cells and T$_{FH}$ cells (CD4$^+$CXCR5$^+$PD-1$^+$) from TC mice using qRT-PCR and normalized to *Ppia*. **b** Hif1α protein expression (as determined by MFI using flow cytometry) in total CD4$^+$ T cells and T$_{FH}$ cells from B6 and TC mice. **c–d** ECAR during a mitochondrial stress test conducted on T$_{FH}$ (CD4$^+$CD44$^+$PD-1$^{hi}$PSGL-1$^{lo}$) and CD44$^-$ CD4$^+$ T cells from B6 and TC mice. **c** Time-course with the arrows indicating the addition of oligomycin (Olig), trifluoromethoxy carbonylcyanide phenylhydrazone (FCCP), and actimycin A and rotenone (Act & Rot). The horizontal gray bar indicates the three time-point measurements of basal ECAR, and the white bar indicates maximum ECAR. The TC and B6 T$_{FH}$ cells plots were significantly different (two-way ANOVA, *P* < 0.001). **d** Basal and max ECAR averages in B6 and TC T$_{FH}$ cells. **e** 2NDBG uptake by B6 and TC T$_N$ cells (CD4$^+$CD44$^-$), T$_{Act}$ cells (CD4$^+$CD44$^+$), and T$_{FH}$ cells (CD4$^+$CD44$^+$PD-1$^{hi}$PSGL-1$^{lo}$). **f** Microarray heat-map and scatter plot of solute transporter genes differentially expressed between B6 and TC T$_{FH}$ cells. Heatmap coloring is mean centered (white), standard deviation normalized to 1 with red indicating above average expression and blue for below average expression. Dotplot coloring is based on signal intensity RMA normalized values, blue is lowest expression and red is highest expression. Selected genes are labeled and only the range 3–9 is shown on dotplot. **g** Expression of solute transporter genes in T$_{FH}$ cells was analyzed using qRT-PCR, and normalized to *Hmbs*. **h** Representative FACS plot (left) and MFI (right) of CD98 expression on TC and B6 T$_{FH}$ cells. All T cells were isolated from spleens from 7–8 month-old mice. Mean + s.e.m. of N = 4–7 per strain, compared with *t* tests (**a**, **b**, **d**, **e**, **g**, **h**), *P < 0.05, **P < 0.01, and ***P < 0.001

CD44$^-$CD4$^+$ T cells were similar between the two strains (Fig. 5c, d). Oxygen consumption rate (OCR) was similar between B6 and TC T$_{FH}$ cells, (Supplementary Fig. 5b). These results confirm that autoreactive T$_{FH}$ cells have a high metabolic demand fueled through glycolysis. Glucose uptake was similar between B6 and TC T$_{FH}$ cells (Fig. 5e) as we have previously shown for total CD4$^+$ T cells[20], suggesting that glucose utilization rather than uptake distinguishes spontaneous lupus T$_{FH}$ cells. We next compared gene expression between TC and B6 spontaneous T$_{FH}$

cells. Among ~1100 differently expressed genes (Supplementary Fig. 5c, d), the differentially expressed pathways confirmed the immunophenotyping results, with greater mTOR activation, proliferation and reduced oxidative phosphorylation in TC T$_{FH}$ cells (Supplementary Fig. 5e). Although pSTAT3 levels were decreased in TC T$_{FH}$ cells (Supplementary Fig. 1h, i), the IL-6_JAK_STAT3 signaling pathway was overexpressed in TC T$_{FH}$ cells, suggesting that it contributes to their expansion. Finally, the KRAS pathway was also overexpressed in TC T$_{FH}$ cells.

Interestingly, monogenic mutations in this pathway are associated with SLE[38], and KRAS is involved in CD4[+] T cell activation in patients with rheumatoid arthritis[39]. Among the top 10 differentially expressed genes (Supplementary Fig. 5f), we noticed three members of the solute carrier family, including the neutral amino acid transporter *Slc1a5* (*Asct2*), a crucial mediator of TCR-stimulated glutamine uptake in CD4[+] T cells[40], that is necessary for $T_H1$ and $T_H17$ cell differentiation[41]. A striking signature of 31 solute transporter genes characterized spontaneous lupus $T_{FH}$ cells (Fig. 5f and Supplementary Table 1). qRT-PCR analysis confirmed an increased expression of two glycolytic genes, *Slc2a6* (*Glut6*) and *Slc37a2* encoding for glucose-6-phosphate transporter in TC $T_{FH}$ cells (Fig. 5g). We also confirmed the elevated expression of two amino acid transporters *Slc1a4* (*Asct1*) and *Slc16a10* (*Mct10*), and the reduced expression of the amino acid transporter *Slc7a10* (*Asc-1*) in TC $T_{FH}$ cells (Fig. 5g). Slc3a2 (CD98) heterodimerizes with Slc7a5 (Lat1), Slc7a6, Slc7a7, or Slc7a8 (Lat2) to transport hydrophobic large neutral amino acids[42,43] and kynurenine[44]. TC $T_{FH}$ cells showed a reduced expression of CD98 (Fig. 5h), although the gene expression of neither *Slc3a2* nor *Slc7a5* showed a difference in the microarray analysis. Taken together, these results suggest that autoreactive $T_{FH}$ cells have different metabolic requirements for amino acids and in the glucose pathway.

**Spontaneous and Ag-specific $T_{FH}$ cells require glutamine**. To further compare gene expression between TC and B6 $T_{FH}$ cells, we selected a NanoString panel of 100 genes related to $T_{FH}$ cell differentiation, CD4[+] T cell function and metabolism (Supplementary Table 2) to probe a new set of spontaneous TC and B6 $T_{FH}$ cells. We identified a signature of 10 genes that differentiated TC from B6 spontaneous $T_{FH}$ cells in both the microarray and Nanostring results (Fig. 6a). *Slc1a5* expression was reduced in TC spontaneous $T_{FH}$ cells (Fig. 6a, b) and NP-Tet[neg] TC $T_{FH}$ cells (Fig. 6c). Importantly, B6 and TC influenza virus-specific NP-Tet[+] $T_{FH}$ cells showed a similar level of *Slc1a5* expression (Fig. 6c). CD98 expression was also similar in NP-Tet[pos] TC and B6 $T_{FH}$ cells, but lower in TC than B6 NP-Tet[neg] $T_{FH}$ cells (Fig. 6d). Slc1a5 is the major glutamine transporter, and CD98 also acts as a glutamine transporter in TCR-stimulated CD4[+] T cells. Since *Slc7a5* deletion in CD4[+] T cells results in a defective TD-response[33], we hypothesized that exogenous Ag-induced $T_{FH}$ cells require glutamine metabolism. We thus tested whether blocking glutaminolysis with DON inhibited the humoral response to NP-KLH primary immunization (Fig. 7a). DON had no effect on the frequency and number of total CD4[+] T cells in immunized B6 and TC mice (Supplementary Fig. 6a). However, DON reduced the frequency of $T_{EM}$ cells in immunized mice from both strains, and the frequency of CD69[+]CD4[+] T cells in immunized TC mice (Supplementary Fig. 6b, c). Neither the frequency or number of $T_{FH}$ cells, or the $T_{FH}/T_{FR}$ ratio were affected by DON treatment (Fig. 7b, c). However, the expression of Bcl6, CD40L and ICOS was decreased in the $T_{FH}$ cells of DON-treated mice in both strains (Fig. 7d). This suggests that glutaminolysis is required for optimal Tfh cell function and interaction with GC B cells. In addition, DON reduced the frequency of $T_H17$ and $T_H1$ cells but not the frequency of IL-10-producing CD4[+] T cells in immunized TC and B6 mice (Supplementary Fig. 6d), which is consistent with results obtained with *Sle1a5*-deficient T cells[43]. DON treatment of immunized mice had little effect on total B cells (Supplementary Fig. 6e), but it decreased the frequency and number of GC B cells (Supplementary Fig. 6f), as well as the size of GCs and their GL7 expression (Fig. 7e and Supplementary Fig. 6g). The few GCs found in the spleens of DON-treated mice were very small with very few B cells. Accordingly, the frequency and number of Ag-specific GC B cells (Fig. 7f), although

not plasma cells (Fig. 7g), were severely reduced by DON treatment. Accordingly, levels of NP-specific IgG1 and IgM were lower in DON-treated B6 and TC mice (Fig. 7h). The frequency of total plasma cells was reduced by DON only in TD-immunized TC mice (Supplementary Fig. 6h). Furthermore, DON reduced the level of total serum IgM in TD-immunized mice from both strains, but reduced the level of total IgG only in TD-immunized B6 mice (Fig. 7i). The serum of TD-immunized TC mice contains a mixture of autoantibodies and NP-KLH-induced IgG, while the serum of B6-immunized mice contains a greater relative amount of anti-NP-KLH antibodies. Taken together, these results strongly suggest that exogenous Ag-specific GC responses require glutamine metabolism.

GCs form 4–5 d after antigen activation and GC maturation is fully achieved at 7 d[45]. Therefore, we treated B6 and TC mice with DON at 5 and 7 d after NP-KLH immunization (Supplementary Fig. 7a) to test the proximal effect of glutaminolysis inhibition on GC development. DON reduced the frequency but not the number of total B cells in immunized mice from both strains; however, the frequency and number of total and Ag-specific GC B cells or plasma cells were not affected by the treatment (Supplementary Fig. 7b-e). Similar results were obtained with total CD4[+] T cells and $T_{FH}$ cells (Supplementary Fig. 7f, g). As with the continuous DON treatment during immunization (Fig. 7d), Bcl6 expression was decreased in $T_{FH}$ cells in DON-treated B6 mice and there was a trend for TC mice (Supplementary Fig. 7h). Despite this modest effect, if any, of targeted DON treatment on either GC B cells or $T_{FH}$ cells, the production of NP-specific IgG1 and IgM (Supplementary Fig. 7i) was markedly decreased in DON-treated mice from both strains.

Finally, we investigated the effect of a 2-week-long DON treatment on spontaneous GCs formation and anti-dsDNA IgG secretion in 7–8-month-old B6 and TC mice. DON reduced the frequency and number of TC B cells and GC B cells in both strains (Fig. 8a, b), but it did not affect splenic plasma cells (Fig. 8c). The size, number and GL7-staining intensity of GCs were greatly reduced by DON in both stains (Fig. 8d, e). DON treatment also normalized the size of the follicles in TC mice (Fig. 8d, f), which corresponds to a combined decreased number of B cells (Fig. 8a) and CD4+ T cells (Fig. 8g). The frequency $T_{FH}$ cells was not affected by DON, but the number of $T_{FH}$ cells was reduced to B6 levels in DON treated TC mice (Fig. 8h), and they expressed a lower level of Bcl6 (Fig. 8i). Finally, the production of anti-dsDNA IgG was reduced in both strains (Fig. 8j), although from a much lower level in B6 mice. The potential therapeutic effect of DON on renal pathology could not be examined due to the short duration of the treatment. Taken together, these results indicate that both Ag-specific and spontaneous GC formation as well as the resulting antigen-specific Ab and autoantibodies are sensitive to the inhibition of glutamine metabolism. Similarly, both antigen-specific and spontaneous GC B cells and $T_{FH}$ cells require glutamine, although to a lesser extend for $T_{FH}$ cells.

## Discussion

We show here that the expansion of $T_{FH}$ cells reported in SLE patients and in lupus-prone mice[4] occurs early before disease manifestation. The frequency of $T_{FH}$ cells is twice higher in pre-autoimmune TC mice as compared to B6 mice, further supporting a causative role. Therapeutic targeting of $T_{FH}$ cells has been proposed for SLE patients[46,47]. In addition to their direct effect on autoantibody production, IL-21 impairs human Treg cells from SLE patients in association with mTOR activation[48]. Therefore reducing the number of lupus $T_{FH}$ cells may have the added benefit of enhancing Treg number and functions. The mechanisms responsible for the expansion of $T_{FH}$ cells in lupus

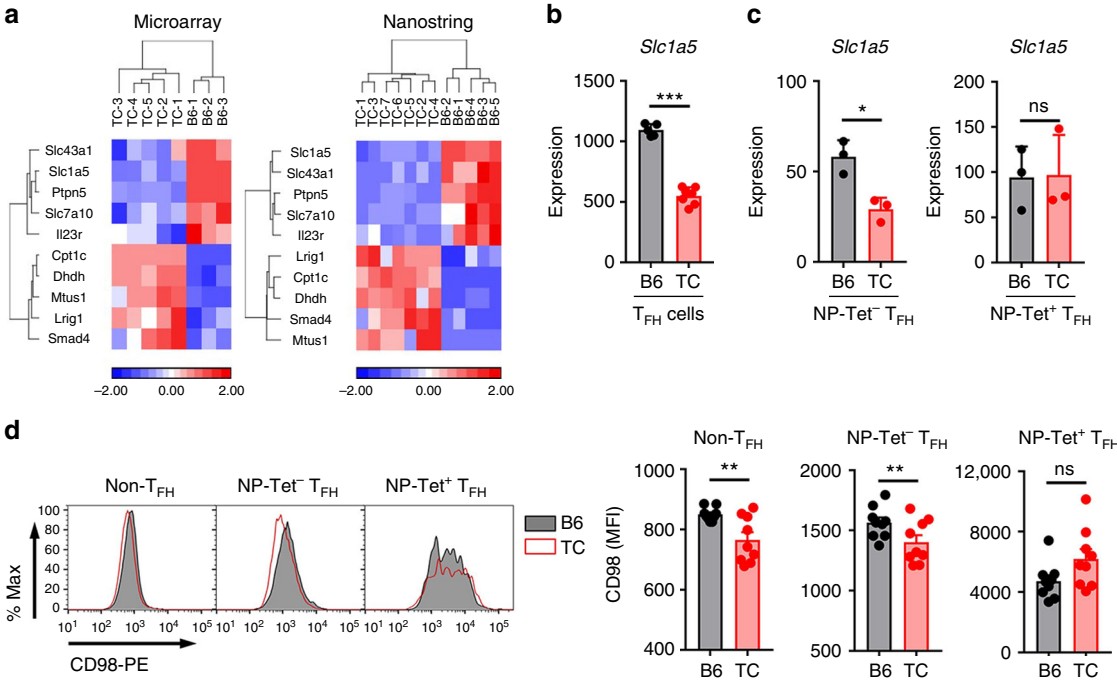

**Fig. 6** Spontaneous but not virus-specific TC $T_{FH}$ cells expressed lower levels of glutamine transporters. **a** Gene signature of spontaneous CD4+CXCR5+PD-1+ $T_{FH}$ cells using microarray (left) and Nanostring (right) on two independent cohorts of TC and B6 mice. **b**, **c** *Slc1a5* expression was analyzed with the Nanostring panel in spontaneous $T_{FH}$ cells (**b**), as well as in influenza virus-specific NP+ $T_{FH}$ cells (CD4+CXCR5+PD-1+NP-Tet$^{pos}$, right) versus non-specific $T_{FH}$ cells (CD4+CXCR5+PD-1+NP-Tet−, left) (**c**). **d** Representative FACS plot (left) and MFI (right) of CD98 expression on influenza virus-specific NP-Tet$^{pos}$ $T_{FH}$ cells (CD4+CXCR5+PD-1+NP-Tet$^{pos}$) and non-specific $T_{FH}$ cells (CD4+CXCR5+PD-1+NP-Tet$^{neg}$), as well as non-$T_{FH}$ (CD4+CXCXR5−PD1−) cells. All T cells were isolated from spleens. Mean + s.e.m. of $N = 3$–9 mice per group compared with $t$ tests, *$P < 0.05$, **$P < 0.01$, and ***$P < 0.001$

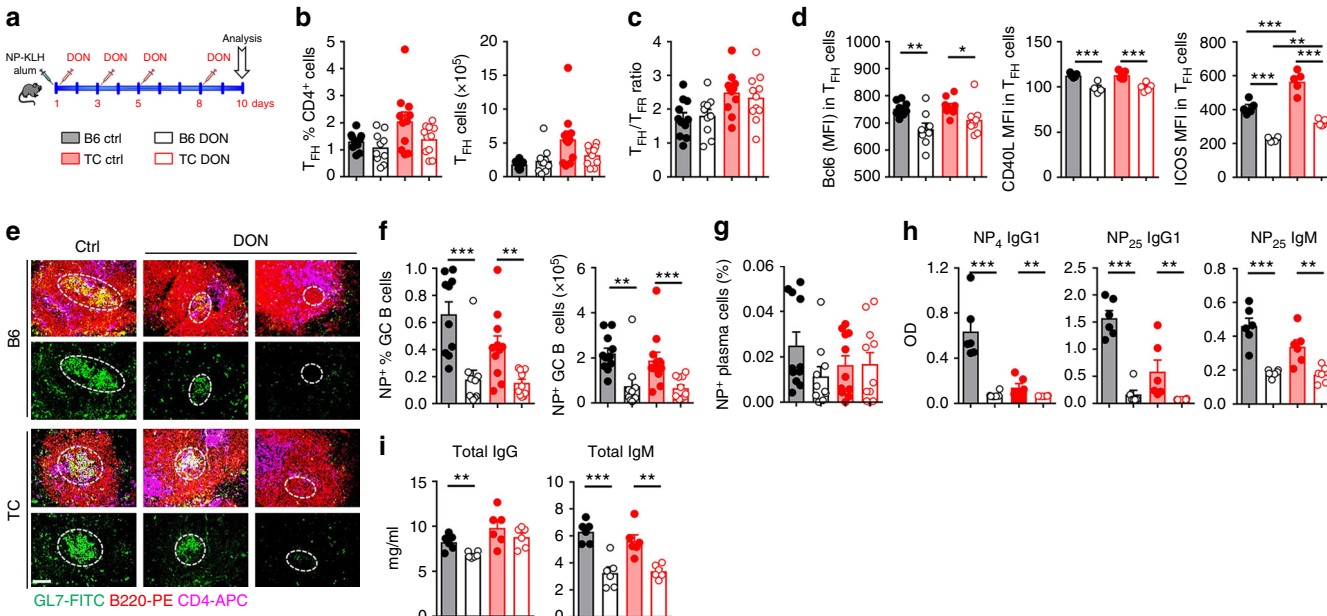

**Fig. 7** Glutamine metabolism is required for both induced and spontaneous antibody responses. **a** Experimental design for immunization with NP-KLH in alum in 8–10-week-old B6 and TC mice treated or not with glutamine antagonist DON. Frequency and cell number of $T_{FH}$ cells (**b**), and ratio of $T_{FH}$ to $T_{FR}$ cells (CD4+CXCR5+PD1+Bcl6+FOXP3− and CD4+CXCR5+PD1+Bcl6+FOXP3+) (**c**). **d** MFI of Bcl6, CD40L and ICOS in $T_{FH}$ cells. **e** Representative spleen sections with GCs shown in the dotted lines. The first rows show composite images and the second rows show GL7-staining only. All images are 10× amplification (scale bar: 200 μM). **f**, **g** Frequency and number of NP-specific GC B cells (**f**) and number of NP-specific plasma cells (**g**). Serum levels of high-affinity anti-NP$_4$, low-affinity anti-NP$_{25}$ IgG1, and anti-NP$_{25}$ IgM NP-specific antibodies (**h**), and total IgG and IgM (**i**). All cell analyses were performed with splenocytes. Mean + s.e.m. of $N = 4$–12 mice per group compared with $t$ tests, *$P < 0.05$, **$P < 0.01$, and ***$P < 0.001$

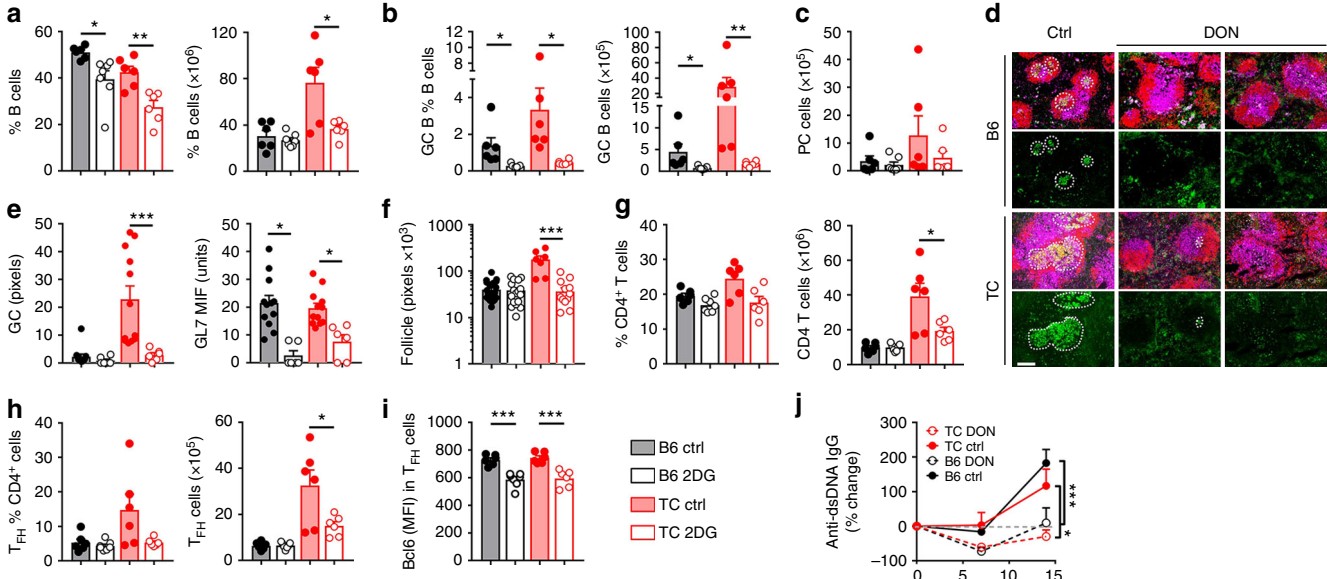

**Fig. 8** Effect of DON treatment on spontaneous GC formation. **a–c** B6 and TC mice were treated with DON 3 times a week for 2 weeks starting at 7–8 months of age. Frequency and number of total B cells (**a**), GC B cells (**b**) and plasma cells (**c**). **d** Representative spleen sections stained with B220-PE (red), CD4-APC (purple) and GL7-FITC (green). The first rows show composite images and the second rows show GL7-staining only. GCs are shown in the dotted lines. All images are 10× amplification (scale bar: 200 μM). **e**, f GC (**e**) and follicle (**f**) surface area as measured by GL7+ pixels (e, left) and GL7 intensity in these areas (f, right), and by B220+ pixels in 3–5 high power fields from 2 mice per group. **g–i** Frequency and number of total CD4+ T cells (**g**), $T_{FH}$ cells (**h**), and MFI of Bcl6 in $T_{FH}$ cells (**i**). **j** Serum anti-dsDNA IgG represented as percent change from pre-treatment value. Statistical comparisons were made between terminal samples. All cell analyses were performed with splenocytes. Mean + s.e.m. N = 6 mice per group compared with t tests, *P < 0.05, **P < 0.01, and ***P < 0.001

are poorly understood[46]. We show here that the expansion of $T_{FH}$ cells in lupus-prone TC mice is associated with an increased expression of Bcl2. Proliferation and STAT3 signaling were similar between TC and B6 $T_{FH}$ cells, but increased in naive TC CD4+ T cells, in which glycolysis is also higher than in naive B6 CD4+ T cells[20]. These findings suggest a model in which the accumulation of $T_{FH}$ cells in TC mice results from the combination of an accelerated induction in proliferating glycolytic naive CD4+ T cells in which STAT3 is activated, possibly in response to the high level of IL-6 present in these mice[49–51], and a resistance to apoptosis.

Immune cells, and T cells in particular, are functionally regulated by their metabolic substrate utilization[52], and alterations in the metabolism of immune cells have been reported in lupus[17,18]. A combination of two metabolic inhibitors, metformin, which targets mitochondrial respiration, and 2DG, an inhibitory glucose analog, has therapeutic effects in vivo in mice and in vitro with human SLE CD4+ T cells[20,21]. In this study focused on Tfh cells instead of clinical outcomes, we show that inhibiting glycolysis is sufficient to reverse the expansion of $T_{FH}$ cells in four different models of lupus, with a concomitant reduction of GC B cell expansion and anti-dsDNA IgG production. Glucose inhibition reduces mTORC1 activation, which may be responsible for the reduction in Bcl6 levels in these $T_{FH}$ cells. The dramatic response of lupus $T_{FH}$ cells to 2DG while reducing the frequency of $T_{EM}$ cells requires the combination of 2DG and metformin ([21] and this study) indicate that lupus $T_{FH}$ cells have a greater glucose requirement than other activated effector CD4+ T cells. A recently completed clinical trial for sirolimus in SLE patients showed that mTOR inhibition decreased the frequency of peripheral CD8+ $T_{EM}$ cells in responsive patients[53]. Since mTOR activation and glycolysis are tightly linked, it is likely that glycolysis contributes at least in part to the expansion of these CD8+

$T_{EM}$ cells, and their novel predictive value warrants further examination in lupus mice.

An effective and safe therapy in SLE should target the $T_{FH}$ cells that drive the production of pathogenic autoantibodies without affecting $T_{FH}$ cells responding to pathogens. We used two different models to address the impact of glucose inhibition on $T_{FH}$ cells induced by exogenous antigens. NP-KLH immunization does not address directly the glucose requirements of Ag-specific $T_{FH}$ cells, but class-switched high affinity NP-antibodies are clearly $T_{FH}$-dependent. In both primary and memory responses, glucose inhibition does not impair the generation of NP-specific GC B cells and anti-NP IgG, while it eliminates anti-dsDNA IgG in the same mice. The influenza virus infection model confirms that the induction of Ag-specific $T_{FH}$ cells is not impaired by 2DG, and that 2DG-treated $T_{FH}$ cells fully support the production of flu-specific antibodies, while again, in the same mice, anti-dsDNA IgG levels are reduced. Therefore, these results suggest that glucose inhibition can eliminate autoreactive $T_{FH}$ cells while preserving the ability to mount a protective response to pathogens. Selective targeting of pathogenic autoreactive immune cells has been an elusive goal in lupus and other autoimmune diseases. Our results suggest that this could be achieved though the targeting of T cell metabolism.

We propose a model in which all $T_{FH}$ cells, especially those induced by exogenous antigens, require glutamine and spontaneous autoreactive $T_{FH}$ cells are uniquely glycolytic (Supplementary Fig. 8). This model is supported by the differential expression of solute transporters favoring amino acid flux in spontaneous B6 $T_{FH}$ cells and glucose flux in TC $T_{FH}$ cells. Moreover, the expression of amino acid transporters *Slc1a5* and CD98 is reduced in spontaneous TC $T_{FH}$ cells but not in virus-specific $T_{FH}$ cells. Finally, the glutamine inhibitor DON prevents the production of NP-specific IgG, but maintains total IgG levels

in TC mice, suggesting a preferential requirement for glutamine in Ag-induced $T_{FH}$ cells. In addition to being an effective T cell inhibitor in alloreactive responses[54], we show here that DON inhibits Ag-specific $T_{FH}$ cells. mTORC1 is activated in T cells either by TCR/PI3K/AKT signaling or amino acid uptake[55], specifically through CD98[33]. The low CD98 expression in spontaneous TC $T_{FH}$ cells suggests that their high mTORC1 activation is fueled by the PI3K/AKT axis, which in turn enhances glycolysis. On the other hand, our results suggest that CD98 mediated mTORC1 activation is required for the induction of Ag-specific $T_{FH}$ cells. mTORC1 activation is necessary in $T_{FH}$ cells[25,26], and it promotes Bcl6 translation in autoreactive $T_{FH}$ cells[27], which is consistent with high mTORC1 activation in spontaneous lupus $T_{FH}$ cells. PR8 virus-induced $T_{FH}$ cells present however a similar level of mTORC1 between lupus and control mice, which is consistent with the relatively low level of mTORC1 activation that was reported in LCMV-induced $T_{FH}$ cells[22]. The elucidation of the mechanisms of mTORC1 activation in autoreactive spontaneous vs. Ag-induced $T_{FH}$ cells will be critical to understand in their function.

The different metabolic requirements of spontaneous autoreactive and Ag-induced $T_{FH}$ cells imply that these two types of $T_{FH}$ cells are qualitatively different, either in their differentiation or function. Spontaneous GC B cells have unique intrinsic requirements for IFNγ and IL-6 signaling in two different autoreactive models[56,57]. The differential response of spontaneous and Ag-induced GC B cells to 2DG mirrored the response of the corresponding $T_{FH}$ cells in TC mice. Both spontaneous and induced GC B cells were however sensitive to glutamine inhibition. The development and maintenance of $T_{FH}$ and GC B cells are tightly linked and the treatment with 2DG could affect directly both cell types. Very little is known on GC B cell metabolism, except that, as $T_{FH}$ cells, GC B cells induced upon immunization require intrinsic mTORC1 activation[58,59]. In addition, treatment with rapamycin during influenza infection alters affinity maturation of virus-specific antibodies[60]. mTORC1 is also necessary for optimal antibody synthesis in long-lived plasma cells suggesting diverse roles of mTOR in humoral immunity[61]. Our results suggest either that autoreactive or Ag-induced GC B cells have different metabolic requirements, or that their differential response to 2DG and DON is a consequence of the effect that these inhibitors have on $T_{FH}$ cells. An expansion of circulating $T_{FH}$ cells has been reported in multiple autoimmune diseases besides SLE[62]. Further investigation of the metabolic requirements of $T_{FH}$ and GC B cells in these autoimmune conditions relative to pathogen exposure should advance the quest for targeted therapeutics in autoantibody-mediated autoimmune diseases as well as our understanding of the mechanisms by which autoreactive GCs are generated and maintained.

## Methods

**Mice and treatment of metabolic inhibitors.** C57BL/6J (B6), NZB/W F1, B6.MRL-Fas[lpr]/J (B6.lpr) and BXSB.Yaa mice were originally obtained from the Jackson Laboratory. B6.NZM-Sle1$^{NZM2410/Aeg}$Sle2$^{NZM2410/Aeg}$Sle3$^{NZM2410/Aeg}$/LmoJ (TC) congenic mice have been previously described[19]. All mice were bred and maintained at the University of Florida and Jackson Laboratories (BXSB.Yaa) in specific pathogen-free conditions. For in vivo metabolic inhibitor treatments, 2DG (Sigma, 6 mg/ml) or metformin (Sigma, 3 mg/ml) were dissolved in drinking water, and mice received 2DG alone or the combination of the two drugs for the duration indicated for each study. Age-matched control mice receiving plain drinking water were used as controls. DON (1.6 mg/kg) was administered intra-peritoneally every other day for a maximum of 2 weeks. For spontaneous GC formation, lupus-prone mice were first screened for serum anti-dsDNA IgG levels between 5 and 6 months of age. Anti-dsDNA IgG-positive mice (i.e., >+2 SD than the average for age-matched B6 controls) were assigned to treated or control groups to distribute equally the autoantibody levels. Only female mice at the age indicated for each experiment were used in this study under protocols approved by the Institutional Animal Care and Use Committees of the University of Florida and the Jackson Laboratory.

**Flow cytometry and immufluorescence staining.** Single-cell suspensions were prepared from spleens using standard procedures. Lungs from PR8-virus infected mice were processed with LiberaseTL (Sigma)[63]. After red blood cell lysis, cells were blocked with anti-CD16/32 Ab (2.4G2), and stained in FACS staining buffer (2.5% FBS, 0.05% sodium azide in PBS). Fluorochrome-conjugated Abs used for FASC are listed in Supplementary Table 3. Follicular T cells were stained in a three-step process using purified CXCR5 (2G8) followed by biotinylated anti-rat IgG (Jackson ImmunoResearch Laboratory) then PerCP-labeled streptavidin in FACS staining buffer on ice. For the intracellular staining, cells were fixed and permeabilized with Foxp3 stating buffer (Thermo Fisher) according to the manufacturer's protocol. 4-Hydroxy-3-nitrophenylacetyl (NP)-Phycoerythrin was purchased from Biosearch Technology. Dead cells were excluded with fixable viability dye (eFluor780; Thermo Fisher). Data were collected on LSRFortessa (BD Biosciences) and analyzed with FlowJo software (Tree Star). To measure glucose uptake, splenocytes were stained with subset-specific antibodies, then resuspended in PBS with 1 μM 2-deoxy-2-[(7-nitro-2,1,3-benzoxadiazol-4-yl)amino]-D-glucose (2NBDG; Life Technologies) for 15 min. After twice washing, cells were analyzed for 2NDBG uptake using flow cytometry. Immunofluorescence staining of spleen frozen sections was performed with an mTOR polyclonal antibody (PA5-34663, 1:1000 dilution) followed by an Alexa Fluor 594-conjugated donkey anti-rabbit IgG (A-2107, 1:200 dilution), both from Thermo-Fisher. GCs were stained with anti-GL7-FITC (LY-77; 1:25), B cells with anti-IgD$^b$-PE (217-170, 1:50) and CD4$^+$ T cells with anti-CD4-APC (RM4-5, 1:100), all from BD Bioscience.

**TD Immunization, ELISA, and ELISPOT.** For NP-KLH immunization, 8–10-week-old mice were pretreated 2DG for 2 weeks before receiving an intra-peritoneal injection of a 100 μg NP(31)-KLH (Biosearch Technology) in alum. The primary response was analyzed 10 d after immunization, and the recall response was analyzed 7 weeks after the first immunization after boosting with same Ag at weeks 2 and 6. To analyze glutamine metabolism in the primary TD response, DON (1.6 mg/kg, Sigma-Aldrich) was administered intra-peritoneally every other day starting on the day of immunization with NP-KLH, or at days 5 and 7. Serum NP-specific antibodies were measured by ELISA using plates coated with NP(4)- or NP(25)-BSA (high or low affinity, respectively) (Biosearch Technology), followed by incubation with 1:1000 diluted serum samples, and developed with alkaline phosphatase-conjugated goat anti-mouse IgG1 (1071-04, 1:1000) or IgG2a (1071-04, 1:1000), both from Southern Biotech. All samples were run in duplicate. Anti-dsDNA IgG were measured in sera diluted 1:100 in plates coated with 50 ug/ml dsDNA. Bound IgG was detected using alkaline phosphatase-conjugated anti-mouse IgG (115-055-003, Jackson ImmunoResearch) diluted 1:1000. Relative units were standardized using serial dilutions of a positive serum from TC mice, setting the 1:100 dilution reactivity to 100 U[64]. Spleen antibody-secreting cells were quantified by an NP-specific IgG ELISPOT assay on serially diluted splenocytes using 96-well mixed cellulose esters membrane plates (Millipore) coated with 50 μg/ml NP(25)-BSA[35]. Bound cells were detected with HRP-conjugated-anti-IgG (1036-05, Southern Biotech) at 1:1000 dilution, and developed by 3-amino-9-ethylcarbazole (Sigma-Aldrich). Antibody-secreting cells were then counted and measured using a Bioreader 4000 Pro-x (Bio-Sys). Germinal center imaging was performed on frozen sections from spleens collected 10 d after immunization.

**Influenza virus infection.** Influenza A virus strain A/Puerto Rico/8/1934 H1N1 (PR8) was kindly provided by Paul G. Thomas (St. Jude Children's Research Hospital). Infection with PR8 virus was carried out on 8–10 week old 2two weeks after treatment with 2DG started. A volume of 30 μl of HBSS containing $1 \times 10^2$ PFU (immunization low dose) or $1 \times 10^6$ PFU (challenge high dose) of PR8 virus were inoculated intranasally (i.n.) in mice anesthetized with isoflurane. To detect virus-specific CD4$^+$ T cells and CD8$^+$ T cells, fluorescence-conjugated MHC-II and MHC-I peptide tetramers, respectively, were obtained from the National Institutes of Health Tetramer Core facility (Emory University, Atlanta, GA). After single cell suspensions ($5 \times 10^6$ cells) were stained with CD4$^+$ T cell antibodies, then incubated with 30 μg/ml of influenza A nucleoprotein (NP) peptide (311-325, QVYSLIRPNENPAHK) I-A$^b$ tetramers for 2 h at 37 °C in a 5% $CO_2$ atmosphere. Virus specific CD8$^+$ T cells were detected with NP peptide (366–374, ASNEN-METM) H-2D$^b$ tetramers. To test the protective effect of serum from immunized mice, cohorts of B6 and TC mice were treated or not with 2DG. Two weeks later, all mice were infected i.n. with a low dose of PR8 virus. 3 weeks later, mice were sacrificed, immune sera were collected. 150 μl of 2DG serum or control serum was transferred intra-peritoneally to naive B6 recipients. One day after transfer, mice were infected i.n. with a high dose of PR8 virus. Naive mice that did not receive serum were used as negative controls for protection and mice that were originally immunized intra-nasally were used as positive controls for protection. Animals were weighed daily and mean percent of initial body weight was calculated for each group. Influenza A NP-specific antibodies were measured in serum collected 30 d after low dose immunization by ELISA. Plates were coated with 100 μl of recombinant NP at 1 μg/ml (Novus Biologicals), followed by incubation with serial dilutions of serum samples (50, 100, 200, 400, 800, and 1600) and developed with alkaline phosphatase-conjugated goat anti-mouse IgG1 (1071-04), IgG2b (1091-04), and IgG2c (1078-04, all from Southern Biotech) each at 1:1000 dilution. The optimal serum dilution to show differences between groups was determined to be 1:200. All samples were run in duplicate.

**Gene expression.** $T_{FH}$ cells (CD4$^+$PD-1$^+$CXCR5$^+$) sorted from 8–10-month-old mice were pooled to generate 5 TC and 3 B6 samples. Total RNA was extracted with RNeasy Mini kit (Qiagen). 200 ng of total RNA per sample was used to generate labeled cDNA fragments with GeneChip$^{TM}$ WT cDNA Labeling Kit (Thermo Fisher) that were hybridized to Affymetrix MTA 1.0 microarrays (Thermo Fisher) with Thermo Fisher reagents. Microarrays were processed with the GeneChip 3000 7G scanner and GeneChip Fluidics Station 450 (Thermo Fisher). Raw CEL files were normalized by the RMA algorithm with Partek Genomic Suite 6.6 (Partek). SYBR green (Biorad)-based qPCR was performed on cDNA synthesized from sorted T cell subsets with primer sequences shown in Supplementary Table 4. Normalization to housekeeping genes *Hmbs* or *Ppia* was carried out with the $2^{-\Delta\Delta Ct}$ method. A custom nCounter Gene Expression CodeSet panel (Supplementary Table 2) was generated by NanoString Technology. Using 50 ng of total RNA per sample, cartridge preparation and scanning was carried out according to manufacturer's instructions using the NanoString Prep Station and nCounter Digital Analyzer. Data was analyzed with nSolver Analysis software, version 2.5 with normalization utilizing positive and negative control probes as well as housekeeping genes.

**Statistical analysis.** Differences between groups were evaluated by two-tailed statistics: unpaired *t* tests or Mann–Whitney *U* tests depending on whether the data were normally distributed, and two-way ANOVA tests for time-course experiments. Unless specified, graphs show means and standard error of the mean (s.e.m.).

## Data availability

Microarray data shown in this study have been deposited in the Gene Expression Omnibus (GEO) database with accession GSE106455. The authors declare that all other data supporting the findings of this study are available within the paper and its Supplementary Information files.

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

## Acknowledgements

We thank Dr. Henry Baker (University of Florida) for his help with gene array analysis; and Leilani Zeumer-Spataro, Nathalie Kanda for outstanding technical help. This study is supported by grants from the NIH (R01AI045050 and R01 AI128901) and from the Alliance for Lupus Research (TIL-416522) to L.M.

## Author contributions

S.-C.C., G.A. and L.M. conceived and designed the experiments; S.-C.C., A.A.T., G.A. and H.R.S. performed the experiments; S.-C.C., A.A.T., G.A., T.M.B., D.C.R., S.S.-A., and L.M. analyzed the data; and S.-C.C. and L.M. wrote the paper.

## Additional information

**Competing interests:** The authors declare no competing interests.

