## [Peer Review File · Nature Communications]

Reviewers' comments:

Reviewer #1 (Remarks to the Author):

This paper is based on prior findings implicating follicular helper T (TFH) cells in autoimmune diseases, primarily systemic lupus erythematosus. It is assumed that eliminating TFH cells would compromise the production of protective antibodies against viral pathogens. Here, Choi et al. propose that inhibiting glucose metabolism results in a drastic reduction of the frequency and number of TFH cells in lupus-prone mice. However, this inhibition had little effect on the production of T-cell-dependent antibodies following immunization with an exogenous antigen or on the frequency of virus-specific TFH cells induced by infection with influenza virus. The solute transporter gene signature showed a different glucose and amino acid flux between autoimmune TFH cells and exogenous antigen-specific TFH cells. Glutaminolysis inhibition reduced both immunization induced and autoimmune humoral responses, but spared autoreactive TFH cells. Thus, blocking glucose metabolism provides an effective and safe therapeutic approach for systemic autoimmunity by eliminating autoreactive TFH cells. Unfortunately, there are several concerns which need to be addressed to support several claims in the abstract. Reference to prior literature should be enhanced and discussion of potential novelty needs to be far more thorough.

Specific comments:

Abstract: Expansion of Tfh cells is not a feature of all autoimmune diseases. Although the involvement of Tfh cells in autoantibody-mediated diseases, such as SLE is evident from prior studies, their role in other autoimmune diseases is far less well defined. Therefore, this generalization should be revised with a clear focus on SLE.

Figure 1. Data in this figure describe increased mTORC1 activity and enhanced survival in TFH cells of lupus-prone TC mice. These findings have been documented by several other groups in lupus-prone mice, including the regulation of bcl6 expression by mTORC1 (Immunity. 2015 Oct 20;43(4):690-702. doi: 10.1016/j.immuni.2015.08.017. Nat Commun. 2017 Aug 15;8(1):254. doi: 10.1038/s41467-017-00348-3.). The following papers also implicated Stat3 in mediating the effect of mTORC1 on bcl6 and Tfh development (Immunity. 2017 Sep 19;47(3):538-551.e5. doi: 10.1016/j.immuni.2017.08.011; J Immunol. 2017 Oct 1;199(7):2377-2387. doi: 10.4049/jimmunol.1700106. Epub 2017 Aug 28.). Apparently, some of these other studies are not cited in the paper. They should be cited and discussed in depth within the context of new findings in this study.

Figure 2. This figure describes the role of glycolysis in Tfh development. Such findings have been reported earlier by Shrestha et al. (Nat Immunol. 2015 Feb;16(2):178-87. doi: 10.1038/ni.3076. Epub 2015 Jan 5.), which should be also cited and discussed. Ray et al have implicated a role for glycolysis in Tfh cells of SLE mice (Immunity. 2015 Oct 20;43(4):690-702. doi: 10.1016/j.immuni.2015.08.017. Epub 2015 Sep 22.). Zeng et al had used 2DG to block the activation of Tfh cells (Immunity. 2016 Sep 20; 45(3): 540-554).

Figure 3 describes that glycolysis inhibition does not affect the TD-humoral response. However, Figure 3A shows a profound reduction of anti-DNA production, which is in apparent contrast to findings reported by the same laboratory in Figures S11 in Sci. Transl. Med. (www.sciencetranslationalmedicine.org/cgi/content/full/7/274/274r_a18/DC1). Such discrepancy needs to be discussed.

Figure 5 shows altered expression of solute transporters in TC TFH cells including Mct4 and CD98. These findings are novel and potentially interesting.

Figure 6 is stated to document that glutamine metabolism is required for both induced and spontaneous antibody production. The figure provides no such evidence. Neither glutamine nor its metabolites were measured to support the stated claim. It is important to provide evidence by

independent approaches that glutamine metabolism is required for both induced and spontaneous antibody production.

Figure 7 shows that inhibition of glutaminolysis with the glutamine analog 6-Diazo-5-oxo-L-norleucine (DON) prevented the production of TD antigen specific antibodies in both B6 and TC mouse strains. However, this intervention has very moderate effects on anti-DNA production. Importantly, there are known interventions that exert much more robust effects on anti-DNA in lupus-prone mice.

Reviewer #2 (Remarks to the Author):

B follicular helper T (Tfh) cells are a spatially and transcriptionally distinct subset of CD4+ T cells required for productive, and implicated in pathogenic, B cell responses. Although the pathways required for the development of these cells have been extensively studied in mouse models of infection and autoimmunity, the metabolic requirements of these cells are less well understood. Choi and colleagues, in their manuscript entitled Inhibition of glucose metabolism selectively targets autoreactive follicular helper T cells, demonstrate different metabolic requirements for cells exhibiting a Tfh cell phenotype in lupus prone mouse strains and those generated in response to an exogenous antigen, studies which include analysis of control non-autoimmune mice. Further, the authors show that differences in susceptibility of these two groups of cells to either glucose or glutaminolysis inhibition correlates with the expression of solute transporter gene expression. This work provides important insights into the metabolic requirements of Tfh cells in an acute as well as the chronic autoimmune setting that adds to a conflicting body of literature.

Below are comments that, if addressed, would strengthen the manuscript:

General comments:

1) Show individual data points on bar graphs. It would be preferred that the authors show individual data points representing individual mice on all bar graphs.

Specific comments:

1) For all figures, indicate the tissue of origin for the cells analyzed in the main text and the legend.

2) Show numbers of cells (CD69+, Tfh, Tfr, and Tem) in addition to frequencies in Figure 1, and to the extent possible for other figures in the paper. For example, in certain figures (S Figure 1), percentages of all populations shown are decreased. Are cell numbers down as well? Do other populations increase?

3) Confirm phenotypes of naïve cells in Figure 1. The authors make the claim in figure 1 that naïve cells exhibit a greater frequency of pSTAT3+ and Ki-67+ based solely on CD44- gating. If the authors want to make this conclusion, they should confirm their phenotype using additional markers of naïve cells (CD62L, CD69, etc.). Are these cells increased in mass? Proliferation? I do not think the phenotype of the naïve cells is important to the manuscript, and would suggest simply removing these cells from the figure.

4) Confirm TC Tfh cell phenotype in situ. It would be desirable to confirm that the cells referred to as Tfh cells expressing an enhanced mTOR phenotype in TC mice reside in germinal centers.

5) Show numbers of cells in addition to frequencies in Figure 2A.

6) Show total percentages and numbers of activated cells (CD44+) in the various lupus models with and without 2DG treatment in Figure 2.

7) Show numbers for of cells in figure 3. It appears that, although the antibody titer against exogenous antigen is not changed, the % of Tfh cells may be significantly decreased in B6 mice treated with 2DG. Cell number data would be the most informative information here.

8) The weight loss data in Figure 4 is confusing, and these results are confounded by the many variables effected glycolysis inhibition. This data should be moved to the supplement if it is not to be better explored.

Show the effect of 2DG treatment on Th1 cells in flu infection. This experiment is potentially flawed by administration of glycolysis inhibitor 2 weeks before infection. Presumably, Th1 cells (and effector CD8+ T cells) would be affected by this treatment. This is important to show as a control, and highlights the diversity of effector subsets elicited by an exogenous antigen. Is there altered mortality in the mice? Viral RNA? Presumably, Th1 cells (and effector CD8+ T cells) would be effected by this treatment. This is important to show as a control, and highlights the diversity of effector subsets elicited by an exogenous antigen.

9) Confirm Hif1a and Mct4 protein expression in TC mice using flow cytometry.

10) To demonstrate Tfh-phenotype cells in TC mice are more dependent on glycolysis than their activated CD4 counterparts, Seahorse analysis should be performed. In addition, it would be informative to measure glucose uptake in these cells using 2NBDG.

11) Present a more global analysis of microarray results using a pathway analysis software. If altered glucose metabolism is a defining characteristic of these cells, one would expect a significant enrichment of some metabolic gene set.

12) Characterize Tfh cells generated with DON and 2DG treatment based on additional functional markers including ICOS, CD40L, IL-21, etc.

13) Finally, one quibble about the introduction. The authors present the current status of metabolic profiling of non-autoimmune Tfh cells as "conflicting", citing references 44, 47, and 48. Careful reading of those papers indicate that the experiments were done quite differently (gene knockout before initiation of proliferation, or early after its onset, vs. retroviral manipulation after cell proliferation). So, a more balanced view of those papers in the introduction is in order, as well as noting that other authors have demonstrated the relative effects of metabolic inhibition in non - autoimmune cells, with data that parallel the current findings.

In conclusion, the manuscript provides important insight into the different metabolic requirements of cells displaying the same phenotype responding to different antigen sources. The novel insights gained regarding the resistance of Tfh generated acutely in response exogenous antigen to glycolysis inhibition represent an important addition to the field of Tfh cell biology. My overall concerns about the work are minor.

Reviewer #3 (Remarks to the Author):

How pathogenic Tfh cells in autoimmune disease animals differ from normal Tfh cells remain largely

unknown. Determining unique biological features of pathogenic Tfh cells will be very important and relevant to clinic, as it may provide a strategy specifically targeting pathogenic Tfh cells without compromising normal Tfh response in autoimmune diseases. Choi et al aimed in this study to define the differences in metabolic requirement between pathogenic Tfh cells and exogenous antigen-specific Tfh cells, and showed that pathogenic Tfh cells were sensitive to inhibition of glucose metabolism. Treatment with 2DG, an inhibitor of glycolysis, decreased the frequency of Tfh cells and GC B cells as well as anti-dsDNA IgG in four strains of lupus-prone mice. 2DG also reduced mTORC1 activation, which was upregulated in CD4+ T cells of TC mice. In contrast, 2DG treatment did not alter the generation of Ag-specific Tfh cells or Ag-specific GC B cells in NP-KLH immunized and in influenza virus-infected B6 mice. The finding in the latter model was particularly striking, because 2DG-treated B6 showed a greater body weight loss after influenza infection, indicating that glycolysis was critical for protection. Furthermore, the study shows that Tfh cells in TC mice and B6 mice show differences in the expression of solute transporter genes. Overall, this study convincingly shows that pathogenic Tfh cells in lupus-prone mice are more sensitive to the inhibition of glycolysis than exogenous Ag-specific Tfh cells. This demonstration in four lupus mouse models is excellent. Although the precise molecular mechanism was not shown in the study, this study provides very important insights into the nature of pathogenic Tfh cells. Yet, several points need to be clarified.

Major points:

1. The large difference between the % and the cell number in Fig. 2E indicates two things: 1) TC mice had ~ 10 more CD4+ T cells than B6 mice, and 2) that 2DG dramatically decreased the entire CD4+ T cell population in TC mice. I agree that the data of Fig. S2C-S2E suggest that Tfh cells were relatively more sensitive to the treatment. But the dramatic decrease of CD4+ T cells in the treated mice suggest that glycolysis was globally important for the entire CD4+ T cell populations in TC mice, possibly including naïve. Can the authors show if the 2DG treatment did not affect the frequency of CD69+ CD4+ T cells and TEM in other three lupus-prone mice? If this was the case, the conclusion that pathogenic Tfh cells were more sensitive to 2DG than other CD4+ T cells would be more convincing. If not, I suggest to modify the conclusion accordingly. The number of total CD4+ T cells should be also provided in each strain +/- 2DG treatment.
2. The authors made an effort to claim that B6 Tfh cells favor amino acid flux, whereas TC Tfh cells favor glucose influx, but this is not supported by the data. The DON treatment did not largely affect the number of Tfh cells or Tfh/Tfr ratio in B6 mice (Fig. 6F), despite the fact that the treatment substantially decreased Ab response and GC response (Fig. 6G-H). A minor reduction in Bcl6 MFI is insufficient to conclude the effect of DON on normal Tfh cells.
3. I am not convinced that exogenous GC response was more dependent on glutamate than autoreactive GC response. The DON treatment strongly inhibited B cell response in both B6 and TC mice. The authors' claim is basically based on the observation in Fig. 6L, where there was no statistical decrease of total IgG in the treated TC mice, but this was measured at only one time point, and I suspect that DON treatment eventually decreased total IgG. Together with the point 2 above, I think that the conclusions in the experiments with DON treatment are too speculative. I suggest to soften the conclusions of this part. Rather, the DON treatment could have been used to address if the inhibition of glutaminolysis changes pathogenic Tfh cells in TC mice.

Minor points

1. Y axis label of Fig. 2E right panel is wrong.
2. P8. "2DG reduced anti-dsDNA IgG production in TC mice in the memory phase, ... in the primary response". Anti-dsDNA IgG production would be little to do with immunization in this part, so "in the memory phase" and "in the primary phase" should be changed to the age of mice.

We are grateful for the constructive detailed critiques presented by each of the three reviewers. We have addressed all the concerns they raised, providing novel results and larger sample sizes to strengthen our conclusions. We have also clarified in the text and in this response to reviewers the areas that were not adequately explained or discussed, with the addition of supporting references as needed. We believe that this manuscript now provides a stronger and cleared body of novel results that support our conclusions.

Our responses to the reviewers are given in blue font. Major changes and additions in the text are highlighted in yellow.

Reviewer #1 (Remarks to the Author):

This paper is based on prior findings implicating follicular helper T (TFH) cells in autoimmune diseases, primarily systemic lupus erythematosus. It is assumed that eliminating TFH cells would compromise the production of protective antibodies against viral pathogens. Here, Choi et al. propose that inhibiting glucose metabolism results in a drastic reduction of the frequency and number of TFH cells in lupus-prone mice. However, this inhibition had little effect on the production of T-cell-dependent antibodies following immunization with an exogenous antigen or on the frequency of virus-specific TFH cells induced by infection with influenza virus. The solute transporter gene signature showed a different glucose and amino acid flux between autoimmune TFH cells and exogenous antigen-specific TFH cells. Glutaminolysis inhibition reduced both immunization induced and autoimmune humoral responses, but spared autoreactive TFH cells. Thus, blocking glucose metabolism provides an effective and safe therapeutic approach for systemic autoimmunity by eliminating autoreactive TFH cells. Unfortunately, there are several concerns which need to be addressed to support several claims in the abstract. *Reference to prior literature should be enhanced and discussion of potential novelty needs to be far more thorough.*

Additional references as well as further clarification and discussion of our results are provided, as detailed below in response to the “specific comments” section.

Specific comments:

Abstract: Expansion of Tfh cells is not a feature of all autoimmune diseases. Although the involvement of Tfh cells in autoantibody-mediated diseases, such as SLE is evident from prior studies, their role in other autoimmune diseases is far less well defined. Therefore, this generalization should be revised with a clear focus on SLE.

An expansion of Tfh and cTfh cells has been found in Sjogren syndrome and type 1 diabetes (reviewed in PMID:27588918), in RA patients (PMID:25475240) and RA mouse models (PMID: 27096318), Myasthenia Gravis (PMID: 27543617), and MS (PMID: 29896193 and PMID:29867938). Although there is not an established causality for these cells in all of these diseases, clinical trials targeting IL-21 are ongoing. However, our work is focused on lupus, therefore we have removed references to other autoimmune diseases, except in the discussion (last paragraph).

Figure 1. Data in this figure describe increased mTORC1 activity and enhanced survival in TFH cells of lupus-prone TC mice. These findings have been documented by several other groups in lupus-prone mice, including the regulation of bcl6 expression by mTORC1 (Immunity. 2015 Oct 20;43(4):690-702. doi: 10.1016/j.immuni.2015.08.017. Nat Commun. 2017 Aug 15;8(1):254. doi: 10.1038/s41467-017-00348-3.). The following papers also implicated Stat3 in mediating the effect of mTORC1 on bcl6 and Tfh development (Immunity. 2017 Sep 19;47(3):538-551.e5. doi: 10.1016/j.immuni.2017.08.011; J Immunol. 2017 Oct 1;199(7):2377-2387. doi: 10.4049/jimmunol.1700106. Epub 2017 Aug 28.).

Apparently, some of these other studies are not cited in the paper. They should be cited and discussed in depth within the context of new findings in this study.

The study by Ray et al 2015 compared the metabolism of Th1 and Tfh cells in the context of LCMV infection in non-autoimmune mice. It showed that mTORC1 activation and glycolysis were downregulated in Tfh related to Th1 cells. These were not sorted for virus specificity, but it can be assumed that the majority of them was virus-induced. This study is cited in our paper (reference 44), as well as the Nat Comm 2017 paper from the Pernis group (reference 49) were cited in the introduction along with other papers as part of a general introduction on what is known about mTOR activation in Tfh cells.

We have added in the discussion (next to last paragraph) a few sentences expending the mTOR discussion in the context of our findings.

We mentioned in the text that STAT3 activation is required for Tfh differentiation with citation of references 50 and 51. We have now clarified by adding that it is due at least in part to its induction of Bcl6 expression, and the Read 2017 reference is added (reference 52).

The decreased pSTAT3 levels that we observed in spontaneous TC Tfh cells is consistent with an decreased IL-21 induced pSTAT3 in SLE patients (reference 53). We have also added p. 12 that the transcriptome of the spontaneous TC Tfh cells showed an increased IL-6/JAK/STAT3 pathway, suggesting an involvement that it is not directly reflected by pSTAT3 levels.

The study by Xu 2017 reported that mTORC1 activation in Tregs induces STAT3 phosphorylation and Bcl6 expression, which are required for Tfr cell differentiation and function. Since the link between pSTAT3 and Bcl6 directly in Tfh cells is already in the other references, this later has not been added.

Figure 2. This figure describes the role of glycolysis in Tfh development. Such findings have been reported earlier by Shrestha et al. (Nat Immunol. 2015 Feb;16(2):178-87. doi: 10.1038/ni.3076. Epub 2015 Jan 5.), which should be also cited and discussed. Ray et al have implicated a role for glucolysis in Tfh cells of SLE mice (Immunity. 2015 Oct 20;43(4):690-702. doi: 10.1016/j.immuni.2015.08.017. Epub 2015 Sep 22.). Zeng et al had used 2DG to block the activation of Tfh cells (Immunity. 2016 Sep 20; 45(3): 540–554).

The study by Shresta at al 2015 addressed the role of Pten in Treg and showed that Tfh cells (and Th1) accumulate in Ptenfl/fl Foxp3-Cre mice as a consequence of Treg dysfunction, including increased glycolysis. The metabolic requirements of Tfh cells were not investigated in this paper.

As stated above, the study by Ray et al 2015 compared the metabolism of Th1 and Tfh cells in the context of LCMV infection in non-autoimmune mice, and it is cited in our manuscript. This study did not mention lupus and showed that LCMV-induced Tfh cells have decreased glycolysis and compared to Th1 cells.

Figure 3 describes that glycolysis inhibition does not affect the TD-humoral response. However, Figure 3A shows a profound reduction of anti-DNA production, which is in apparent contrast to findings reported by the same laboratory in Figures S11 in Sci. Transl. Med. (www.sciencetranslationalmedicine.org/cgi/content/full/7/274/274r_a18/DC1). Such discrepancy needs to be discussed.

We thank the reviewer for having spotted this apparent discrepancy. We have indeed reported that 2DG treatment failed to reverse disease in mice that were treated starting at 7 months of age (Yin 2015, Fig. S11), although there was a trend for a decreased level of anti-dsDNA IgG. We have reported later that treatment with 2DG alone was sufficient to prevent disease and anti-dsDNA IgG production (Yin 2016). The mice used in Fig. 2 were younger, about 5 months of age at the beginning of the treatment, when they became anti-dsDNA IgG positive. Mice used in Fig. 3, 4, 6 and 7 were younger (2 months of age) because older lupus mice respond poorly to TD-immunization when the autoAbs production dominates, and using older mice would have made it difficult to compare the two types of response in the same mouse. The immunophenotyping of Tfh cells in Fig. 1 and the gene expression in sorted Tfh

cells in Fig. 5 were older to provide larger number of cells. Finally, mice in Fig. 8 to evaluate the effect of DON on spontaneous Tfh cells and anti-dsDNA IgG were also older (same age as in the 2015 2DG treatment in Fig. S11), and DON was able to reverse anti-dsDNA IgG when 2DG was not. The age of the mice used in each experiment and the rationale for it is now clearly presented.

Figure 5 shows altered expression of solute transporters in TC TFH cells including Mct4 and CD98. These findings are novel and potentially interesting. Thank you for your interest.

Figure 6 is stated to document that glutamine metabolism is required for both induced and spontaneous antibody production. The figure provides no such evidence. Neither glutamine nor its metabolites were measured to support the stated claim. It is important to provide evidence by independent approaches that glutamine metabolism is required for both induced and spontaneous antibody production. DON has been used to inhibit glutaminolysis in multiple studies, including by the Powell group in a study that is cited in our paper. It has probably some off-target effects and we agree that we have not formally showed that glutamine is required. We have shown that when we use a glutaminolysis inhibitor, we inhibit autoantibody and induced antibody production. We have rephrased our conclusions to reflect this valid nuance.

Figure 7 shows that inhibition of glutaminolysis with the glutamine analog 6-Diazo-5-oxo-L-norleucine (DON) prevented the production of TD antigen specific antibodies in both B6 and TC mouse strains. However, this intervention has very moderate effects on anti-DNA production. Importantly, there are known interventions that exert much more robust effects on anti-DNA in lupus-prone mice. We are trying to convey the point that the inhibition of glycolysis with 2DG inhibits anti-dsDNA IgG but not TD-responses while glutaminolysis inhibition with DON inhibits both. It appears that the effect of DON on TD-antibodies is stronger than on anti-dsDNA IgG, but this is a difficult direct comparison to make, and we are not going further that it affects both. We are not proposing DON (or other drug targeting glutamine metabolism) as a treatment to reduced autoantibody production. We used this reagent as a tool to probe glutamine vs. glucose utilization relative to antibody production.

Reviewer #2 (Remarks to the Author):

B follicular helper T (Tfh) cells are a spatially and transcriptionally distinct subset of CD4+ T cells required for productive, and implicated in pathogenic, B cell responses. Although the pathways required for the development of these cells have been extensively studied in mouse models of infection and autoimmunity, the metabolic requirements of these cells are less well understood. Choi and colleagues, in their manuscript entitled Inhibition of glucose metabolism selectively targets autoreactive follicular helper T cells, demonstrate different metabolic requirements for cells exhibiting a Tfh cell phenotype in lupus prone mouse strains and those generated in response to an exogenous antigen, studies which include analysis of control non-autoimmune mice. Further, the authors show that differences in susceptibility of these two groups of cells to either glucose or glutaminolysis inhibition correlates with the expression of solute transporter gene expression. This work provides important insights into the metabolic requirements of Tfh cells in an acute as well as the chronic autoimmune setting that adds to a conflicting body of literature.

Below are comments that, if addressed, would strengthen the manuscript:

General comments:

1) Show individual data points on bar graphs. It would be preferred that the authors show individual data points representing individual mice on all bar graphs.

We corrected all figures following “Author Instruction” of the journal.

Specific comments:

1) For all figures, indicate the tissue of origin for the cells analyzed in the main text and the legend.

All data are shown from spleen, as indicated in the Material and Methods section. This has also been added to each figure legend. Some cells from the lungs of flu infected mice were also analyzed as indicated in the main text and figure legends.

2) Show numbers of cells (CD69+, Tfh, Tfr, and Tem) in addition to frequencies in Figure 1, and to the extent possible for other figures in the paper. For example, in certain figures (S Figure 1), percentages of all populations shown are decreased. Are cell numbers down as well? Do other populations increase?

We added cell numbers in Figure 1 and Figure S1. Cell numbers showed the same trend with frequencies except Figure S1g. Cell numbers were also added to other figures.

3) Confirm phenotypes of naïve cells in Figure 1. The authors make the claim in figure 1 that naïve cells exhibit a greater frequency of pSTAT3+ and Ki-67+ based solely on CD44- gating. If the authors want to make this conclusion, they should confirm their phenotype using additional markers of naïve cells (CD62L, CD69, etc.). Are these cells increased in mass? Proliferation? I do not think the phenotype of the naïve cells is important to the manuscript, and would suggest simply removing these cells from the figure.

A majority of CD44(-) cells were CD62L(+)/CD69(-) cells, however the frequency of CD44-CD62L+CD69- was lower, and reciprocally, the frequency of CD62L-CD69+ cells was higher in the CD44-CD4+ T cells from aged TC mice, as shown in the figure below (Fig. R1). A higher frequency of CD44-CD4+ T cells in TC mice also expressed pSTAT3 and Ki-67 in both aged and young mice (Fig. 1f and S1g-h). We think this indicates an intermediate activation status in TC CD4+ T cells that exists before disease onset. We are in the process of fully characterizing these cells in the context of Tfh differentiation. We have modified the text accordingly (p. 6)

Fig. R1. Frequency of CD62L+CD69- and CD62L-CD69+ in CD44-CD4+ T cells from aged B6 and TC mice.

4) Confirm TC Tfh cell phenotype in situ. It would be desirable to confirm that the cells referred to as Tfh cells expressing an enhanced mTOR phenotype in TC mice reside in germinal centers.

We have performed immunofluorescence staining for mTOR in spontaneous GCs from aged TC and B6 mice, which confirmed the FACS staining of mTOR expression in the CD4+ T cells in the GCs of TC but not B6 mice. This finding has been added as Fig. 1e and Fig S1f. To make room for this data in Fig. 1, the pSTAT3 results for aged mice have been moved to Fig. S1g with the results for young mice.

5) Show numbers of cells in addition to frequencies in Figure 2A.

We added numbers of Tfh cells and GC B cells as Figure 2b.

6) Show total percentages and numbers of activated cells (CD44+) in the various lupus models with and without 2DG treatment in Figure 2.

This has been added as Sup. Fig. 2g. This data shows that the effect of 2DG alone on the percentage of CD44+CD4+ T cells is variable among strains, with no significant difference in TC and B6.lpr mice, but a difference in NZBW F1 and BXSB.Yaa mice.

7) Show numbers for of cells in figure 3. It appears that, although the antibody titer against exogenous antigen is not changed, the % of Tfh cells may be significantly decreased in B6 mice treated with 2DG. Cell number data would be the most informative information here.

We added cell numbers to all the results in Figure 3.

8) The weight loss data in Figure 4 is confusing, and these results are confounded by the many variables effected glycolysis inhibition. This data should be moved to the supplement if it is not to be better explored.

We have moved the weight loss figure to Fig. S3a.

Show the effect of 2DG treatment on Th1 cells in flu infection. This experiment is potentially flawed by administration of glycolysis inhibitor 2 weeks before infection. Presumably, Th1 cells (and **effector CD8+ T cells**) would be affected by this treatment. This is important to show as a control, and highlights the diversity of effector subsets elicited by an exogenous antigen. **Is there altered mortality in the mice? Viral RNA?** Presumably, Th1 cells (and effector CD8+ T cells) would be effected by this treatment. This is important to show as a control, and highlights the diversity of effector subsets elicited by an exogenous antigen.

All the mice infected with low sub-lethal doses of PR8 flu were sacrificed at day 10 p.i., because the main purpose of the experiment was to characterize the response of induced Tfh cells to 2DG. Serum was collected at d 30 p.i. when the anti-Flu IgG titers are at their peak (PMID: 18768854) in another cohort of mice to measure the effect of 2DG on anti-PR8 antibodies. At either d 10 or d 30 p.i with this non-lethal dose, there was no mortality in either strain, treated or not.

We have added the results for Th1 cells and NP+ CD8 T cells in the lungs (where they are the most critical for protection) in Fig. 4 and in the spleen in Fig. S3. We have also results for Polymerase acidic protein (PA)-specific CD8+ T cells shown Fig. R2, showing essentially the same results: 2DG has no effect on the Th1 and CD8 early response to flu.

There are interesting strain differences, with 2DG favoring MPEC differentiation in B6 and SLEC in TC mice. We are in the process of putting together a paper comparing in details the flu response between B6 and TC mice, including metabolic inhibitors, in which this difference will be explored on long term effects. This paper will include viral titers. This large amount of data is beyond the scope of this current paper, which is focused on Tfh cells.

Fig. R2. From left to right: Numbers a PA-Tet+ CD8+ T cells in the lung and spleen of B6 and TC mice at d 10 after injection with PR8 flu. Frequency and number of SLEC and MPEC among these PA-Tet+ CD8+ T cells

9) Confirm Hif1a and Mct4 protein expression in TC mice using flow cytometry.

Expression of Hif1- α is increased in Tfh as compared with total CD4+ T cells at the protein level, but there is no difference between TC and B6 mice. This has been added as Fig. 5b. We could not test Mct4 protein expression because there is no available anti-Mct4 Ab for flow cytometry, and the cell number is not sufficient for WB.

10) To demonstrate Tfh-phenotype cells in TC mice are more dependent on glycolysis than their activated CD4 counterparts, seahorse analysis should be performed. In addition, it would be informative to measure glucose uptake in these cells using 2NBDG.

We have performed Seahorse on sorted spontaneous Tfh cells (which has not been done before to our knowledge) as compared to CD44neg CD4+ T cells for both strains, and confirmed an enhanced glycolysis in TC Tfh cells. This has been added as Fig 5b and c. No difference was observed for oxygen consumption, and this results was added in Fig. S4a.

2NDBG uptake was compared between naïve T cells, T_{Act} and T_{FH} cells from aged mice, defined by CD4, CD44, PSGL-1, and PD-1 staining. 2NDBG uptake was lower in naïve T cells than the two other subsets, however, there is no difference between TC and B6 mice. Consistent with this result, Glut1 expression was similar between B6 and TC Tfh cells (message and protein levels, data not shown). This confirmed our earlier results (Yin 2015) showing no difference in glucose uptake between B6 and TC total CD4+ T cells, in spite of an upregulation of glycolytic enzymes. This has been added as Fig. 5e.

11) Present a more global analysis of microarray results using a pathway analysis software. If altered glucose metabolism is a defining characteristic of these cells, one would expect a significant enrichment of some metabolic gene set.

A list of the top differentially expressed pathways is now included in Fig. S4e and f. This data is also discussed in the text.

12) Characterize Tfh cells generated with DON and 2DG treatment based on additional functional markers including ICOS, CD40L, IL-21, etc.

We immunized and DON-treated an additional cohort of mice. The results were included to Fig. 7 and S5, and we added results for ICOS and CD40L Fig. 7d (old Fig 6 and been split in 2). No difference were obtained for IL-23R, IL-21R and PD1 (data not shown).

13) Finally, one quibble about the introduction. The authors present the current status of metabolic profiling of non-autoimmune Tfh cells as “conflicting”, citing references 44, 47, and 48. Careful reading of those papers indicate that the experiments were done quite differently (gene knockout before initiation of proliferation, or early after its onset, vs. retroviral manipulation after cell proliferation). So, a more balanced view of those papers in the introduction is in order, as well as noting that other authors have demonstrated the relative effects of metabolic inhibition in non-autoimmune cells, with data that parallel the current findings.

The text has been revised to reflect these valid points more accurately.

In conclusion, the manuscript provides important insight into the different metabolic requirements of cells displaying the same phenotype responding to different antigen sources. The novel insights gained regarding the resistance of Tfh generated acutely in response exogenous antigen to glycolysis inhibition represent an important addition to the field of Tfh cell biology. My overall concerns about the work are minor.

Reviewer #3 (Remarks to the Author):

How pathogenic Tfh cells in autoimmune disease animals differ from normal Tfh cells remain largely unknown. Determining unique biological features of pathogenic Tfh cells will be very important and relevant to clinic, as it may provide a strategy specifically targeting pathogenic Tfh cells without compromising normal Tfh response in autoimmune diseases. Choi et al aimed in this study to define the differences in metabolic requirement between pathogenic Tfh cells and exogenous antigen-specific Tfh cells, and showed that pathogenic Tfh cells were sensitive to inhibition of glucose metabolism. Treatment with 2DG, an inhibitor of glycolysis, decreased the frequency of Tfh cells and GC B cells as well as anti-dsDNA IgG in four strains of lupus-prone mice. 2DG also reduced mTORC1 activation, which was upregulated in CD4+ T cells of TC mice. In contrast, 2DG treatment did not alter the generation of Ag-specific Tfh cells or Ag-specific GC B cells in NP-KLH immunized and in influenza virus-infected B6 mice. The finding in the latter model was particularly striking, because 2DG-treated B6 showed a greater body weight loss after influenza infection, indicating that glycolysis was critical for protection. Furthermore, the study shows that Tfh cells in TC mice and B6 mice show differences in the expression of solute transporter genes. Overall, this study convincingly shows that pathogenic Tfh cells in lupus-prone mice are more sensitive to the inhibition of glycolysis than exogenous Ag-specific Tfh cells. This demonstration in four lupus mouse models is excellent. Although the precise molecular mechanism was not shown in the study, this study provides very important insights into the nature of pathogenic Tfh cells. Yet, several points need to be clarified.

Major points:

1. The large difference between the % and the cell number in Fig. 2E indicates two things: 1) TC mice had ~x10 more CD4+ T cells than B6 mice, and 2) that 2DG dramatically decreased the entire CD4+ T cell population in TC mice. I agree that the data of Fig. S2C-S2E suggest that Tfh cells were relatively more sensitive to the treatment. But the dramatic decrease of CD4+ T cells in the treated mice suggest that glycolysis was globally important for the entire CD4+ T cell populations in TC mice, possibly including naïve. **Can the authors show if the 2DG treatment did not affect the frequency of CD69+ CD4+ T cells and TEM in other three lupus-prone mice?** If this was the case, the conclusion that pathogenic Tfh cells were more sensitive to 2DG than other CD4+ T cells would be more convincing. If not, I suggest to modify the conclusion accordingly. The number of total CD4+ T cells should be also provided in each strain +/- 2DG treatment.

Indeed 2DG decreased the number and frequency of total CD4+ T cells in all lupus strains. This has been added as Sup. Fig. 2f, along with more data on the effect of 2DG on CD44+ activated T cells, which contain Tfh and other effector subsets (Sup. Fig. 2g), also in response to Rev 2, Q6. We have modified the text accordingly, stressing the uniform reduction of Tfh cells and the more variable response of activated T cells to 2DG in the text. A more definitive comparison would require an extensive analysis of the response to 2DG in T cell subsets in all lupus strains as we have now done for TC mice. A manuscript is in preparation expanding on this topic for BWF1 and BXSB.Yaa (Morel &Roopenian)

2. The authors made an effort to claim that B6 Tfh cells favor amino acid flux, whereas TC Tfh cells favor glucose influx, but this is not supported by the data. The DON treatment did not largely affect the number of Tfh cells or Tfh/Tfr ratio in B6 mice (Fig. 6F), despite the fact that the treatment substantially decreased Ab response and GC response (Fig. 6G-H). A minor reduction in Bcl6 MFI is insufficient to conclude the effect of DON on normal Tfh cells.

We have added 2 graphs showing a decreased CD40L and ICOS expression on Tfh cells treated with DON. An additional cohort of DON treated immunized mice showed a significant decrease in Bcl6 (Fig. 7d). We have reworded our interpretation in the text to suggest that DON affect Tfh cell function rather than their frequency and number. We have added numbers of Tfh cells in Fig. 8 for the effect of DON on spontaneous GCs.

3. I am not convinced that exogenous GC response was more dependent on glutamate than autoreactive GC response. The DON treatment strongly inhibited B cell response in both B6 and TC mice. The authors' claim is basically based on the observation in Fig. 6L, where there was no statistical decrease of total IgG in the treated TC mice, but this was measured at only one time point, and I suspect that DON treatment eventually decreased total IgG. Together with the point 2 above, I think that **the conclusions in the experiments with DON treatment are too speculative. I suggest to soften the conclusions of this part.** Rather, the DON treatment could have been used to address if the inhibition of glutaminolysis changes pathogenic Tfh cells in TC mice.

We agree and we cannot conclude at this time that DON affects spontaneous more than TD-induced Tfh cells, therefore we limit our conclusions that glutaminolysis is required by all Tfh cells for optimal development. This is in contrast of the results for 2DG that are more clear cut, drastic effects on spontaneous lupus Tfh cells that we did not observed with DON. The text has been modified accordingly.

Minor points:

1. Y axis label of Fig. 2E right panel is wrong.

Thank you for pointing this out. We have corrected it (now Fig. 2f).

2. P8. "2DG reduced anti-dsDNA IgG production in TC mice in the memory phase, in the primary response". Anti-dsDNA IgG production would be little to do with immunization in this part, so "in the memory phase" and "in the primary phase" should be changed to the age of mice.

This has been reworded to eliminate any ambiguity.

Reviewers' comments:

Reviewer #1 (Remarks to the Author):

The manuscript has been improved. There are few concerns that should be addressed.

As stated in the revised discussion of the manuscript, a reduction of the number of TFH cells by blocking the IL-21 pathway has beneficial effects in BXSb.Yaa and NZB/W F1 lupus mice and therapeutic targeting of TFH cells has been proposed for SLE patients. The mechanisms responsible for the expansion of TFH cells in lupus are poorly understood.

Given that mTOR is a key regulator of glycolysis, which is successfully inhibited here to reduce development and IL-21 production of Tfh cells, it would be critical to reconcile the present results in mice with recent human studies showing evidence for the involvement of the mTOR pathway in the development of IL-21-producing T cells in patients with SLE (*Arthritis Rheumatol.* 70(3):427-438). Along these lines, recent human studies also showed that clinically therapeutic mTOR blockade with sirolimus reversed the depletion of effector-memory T cells in SLE patients (*Lancet*, 391:1186-1196). The depletion of effector-memory T cells was predictive of clinical responsiveness among SLE patients. This appears to be in contrast with the expansion of effector-memory T cells in TC mice in this study. The authors should consider whether comparable markers were used to characterize such T cell subsets and whether the lupus-prone TC mice used in this study accurately reflect the phenotype of sirolimus-responsive SLE patients who had arthritis, dermatitis, and vasculitis.

The authors should clearly state in the discussion which clinical manifestations of lupus have responded to 2DG and DON in this study.

Reviewer #2 (Remarks to the Author):

The authors have done a nice job of experimentally addressing the reviewer comments.

Reviewer #3 (Remarks to the Author):

The authors have nicely addressed my previous concerns, and modified the conclusions accordingly. I think that this is a remarkable study which might provide significant insights into the fundamental features of autoreactive Tfh cells in lupus.

I have one minor comment. Fig. S6j and Fig. 7i show the presence of anti-dsDNA IgG in B6 mice at equivalent levels in TC mice. I found these data just confusing, as they are different from the results in Fig. 2a. I understand that the titers were low, but I do not see the significance of the data anyways. I recommend to remove these figures.

Response to the Reviewers' comments:

Reviewer #1 (Remarks to the Author):

The manuscript has been improved. There are few concerns that should be addressed. As stated in the revised discussion of the manuscript, a reduction of the number of TFH cells by locking the IL-21 pathway has beneficial effects in BXSB.Yaa and NZB/W F1 lupus mice and therapeutic targeting of TFH cells has been proposed for SLE patients. The mechanisms responsible for the expansion of TFH cells in lupus are poorly understood. Given that mTOR is a key regulator of glycolysis, which is successfully inhibited here to reduce development and IL-21 production of Tfh cells, it would be critical to reconcile the present results in mice with recent human studies showing evidence for the involvement of the mTOR pathway in the development of IL-21-producing T cells in patients with SLE (Arthritis Rheumatol. 70(3):427-438).

The study by Kato and Perl showed that IL-21 impairs Treg development and function *in vitro*, which is associated with mTOR activation and impaired autophagy. Since IL-21 levels are elevated in SLE patients, this implies that Treg dysfunctions in SLE may be secondary to Tfh cell expansion. **This has been included in the discussion p. 16.** Consequently, the reduction of Tfh cell expansion by 2DG treatment should have a beneficial effect on Treg function.

We reexamined our results obtained in TC mice treated or not with 2DG. As shown in the figure below, 2DG did not change the percentage of Tregs, but it decreased their numbers (due to the overall reduction of lymphocyte and CD4+ T cells numbers). There was no difference for the level of CD25 expression (data not shown), but the ratio of CD62L⁻/CD62L⁺ Treg cells was decreased by the 2DG treatment. CD62L expression is found on Tregs with better suppressive function¹, suggesting that 2DG may improve the defective function of TC Treg cells^{2,3}. We can only speculate at this point that it may be related to a decreased exposure to IL-21. Although as shown in the paper, the Tfr/Tfh ratio was increased by 2DG, the frequency and number of Tfr cells were reduced by 2DG, indicating that glycolysis inhibition does not enhance the recruitment of Tregs to become Tfr cells. Its main effect is to reduce Tfh cell expansion, resulting in a more favorable Tfr/Tfh ratio. Understanding whether these *in vivo* results are direct or indirect on Tregs will require further studies, including using FOXP3-GFP mice. For this reason, and because the focus of the present paper is on Tfh cells, we have opted to NOT

include this data in the paper.

We tested IL-21 intracellular staining with commercial anti-IL-21 Abs as well as with mouse IL-21 chimera protein, but we felt that neither method is reliable. Sorted spontaneous TC Tfh cells express more *Il21* message than B6 Tfh cells as shown on the figure on the right. This suggests that not only Tfh cells are expanded in the lupus mice, but that they produce more IL-21. We are in the process of pooling 2DG-treated mice to sort Tfh cells for RNASeq, which will tell us, among many other things, whether 2DG normalizes *Il21* expression.

We have bred the *Il21*^{VFP} reporter to B6.*Sle1a1* and B6.*Pbx1d*-Tg mice. The *Sle1a1* locus and *Pbx1d* overexpression in CD4⁺ T cells increase Tfh cell differentiation³. We have shown that this IL-21 reporter construct reliably responds to at least some of the factors involved in Tfh cell expansion in TC mice⁴. Our on-

going analysis of Tfh cells in B6.*Sle1a1. Il21^{VFP}*, B6.*Pbx1d-Tg. Il21^{VFP}* and B6.*Il21^{VFP}* mice includes metabolic pathways and their response to 2DG, and will be published separately.

Along these lines, recent human studies also showed that clinically therapeutic mTOR blockade with sirolimus reversed the depletion of effector-memory T cells in SLE patients (Lancet, 391:1186-1196). The depletion of effector-memory T cells was predictive of clinical responsiveness among SLE patients. This appears to be in contrast with the expansion of effector-memory T cells in TC mice in this study. The authors should consider whether comparable markers were used to characterize such T cell subsets and whether the lupus-prone TC mice used in this study accurately reflect the phenotype of sirolimus-responsive SLE patients who had arthritis, dermatitis, and vasculitis.

We thank the reviewer to bringing up this important point. The effector memory T cells examined in our study and that showed a variable / modest response to 2DG were splenic CD4⁺CD44⁺CD62L⁻, while the effector memory T cells predictive of the sirolimus response in the Lancet paper were peripheral blood CD8⁺CD62L⁻CD197⁻. These two cell types may have different metabolic requirements. We also posit that the inhibition of glycolysis by 2DG and the inhibition of mTOR with sirolimus have most likely overlapping effects but distinct effects. **This has been added to the discussion p. 17.** We are in the process of conducting a thorough analysis of CD4⁺ and CD8⁺ effector and effector memory T cells in the TC mice, comparing spontaneous vs. flu-induced, as well as their response to metabolic inhibitors. These results will be published separately.

The authors should clearly state in the discussion which clinical manifestations of lupus have responded to 2DG and DON in this study.

The clinical outcomes of treatments were not part of this study, which was focused on Tfh cells and anti-dsDNA IgG production as proximal autoimmune phenotype. **This has been added p. 15 and 16**

Reviewer #2 (Remarks to the Author):

The authors have done a nice job of experimentally addressing the reviewer comments. We are happy to have addressed all of this reviewer's comments.

Reviewer #3 (Remarks to the Author):

The authors have nicely addressed my previous concerns, and modified the conclusions accordingly. I think that this is a remarkable study which might provide significant insights into the fundamental features of autoreactive Tfh cells in lupus.

I have one minor comment. Fig. S6j and Fig. 7i show the presence of anti-dsDNA IgG in B6 mice at equivalent levels in TC mice. I found these data just confusing, as they are different from the results in Fig. 2a. I understand that the titers were low, but I do not see the significance of the data anyways. I recommend to remove these figures.

We agree with this comment and have removed Figure S6j and figure 7i.

References

1. Ermann, J. *et al.* Only the CD62L⁺ subpopulation of CD4⁺CD25⁺ regulatory T cells protects from lethal acute GVHD. *Blood* **105**, 2220-2226 (2005).
2. Cuda, C. M., Wan, S., Sobel, E. S., Croker, B. P. & Morel, L. Murine lupus susceptibility locus *Sle1a* controls regulatory T cell number and function through multiple mechanisms. *J. Immunol.* **179**, 7439-7447 (2007).
3. Choi, S. C. *et al.* The lupus susceptibility gene *Pbx1* regulates the balance between follicular helper T cell and regulatory T cell differentiation. *J. Immunol.* **197**, 458-469 (2016).

4. Choi, S. C. *et al.* Relative contributions of B cells and dendritic cells from lupus-prone mice to CD4(+) T cell polarization. *J Immunol* **200**, 3087-3099 (2018).

REVIEWERS' COMMENTS:

Reviewer #1 (Remarks to the Author):

The authors have thoughtfully addressed the previous concerns.